# Global patterns and trends of carbon monoxide poisoning: A comprehensive spatiotemporal analysis using joinpoint regression and ARIMA modeling, 1990–2021

**Weiguang Wang**[1*‡], **Yongai Ling**[1], **Xianwei Xiong**[1], **Jiajie Zhou**[2]

**1** Emergency Department, People's Hospital of Anji, Huzhou City, Zhejiang Province, China, **2** Department of Gastrointestinal Surgery, The Affiliated Huaian No.1 People's Hospital of Nanjing Medical University, Huai'an City, Jiangsu Province, China

‡ Weiguang Wang first author.
* batteryweiger2018@163.com

## Abstract

### Background

Carbon monoxide (CO) poisoning causes approximately 41,000 deaths annually worldwide despite being preventable. Previous studies focused primarily on mortality alone, lacked systematic socio-demographic analysis, and provided no predictive models. This study comprehensively analyzes global CO poisoning patterns using spatiotemporal methods to inform evidence-based prevention strategies.

### Methods

We analyzed Global Burden of Disease Study 2021 data from 204 countries (1990–2021) for age-standardized incidence, mortality, and disability-adjusted life years (DALYs). Joinpoint regression identified temporal trends with statistical precision, spatial statistics quantified geographic clustering, and ARIMA modeling projected trends through 2050. We examined associations with socio-demographic index (SDI) across regions and countries.

### Results

Global age-standardized incidence rates decreased significantly by 35.1% from 12.13 (95% UI: 8.30–17.00) to 7.87 (95% UI: 5.54–10.81) per 100,000 population (annual percentage change: −1.16%, 95% UI: −1.35% to −0.96%, p<0.001). Mortality rates declined more dramatically by 53.9% from 0.76 (95% UI: 0.66–0.91) to 0.35 (95% UI: 0.24–0.40) per 100,000 (annual change: −2.79%, 95% UI: −3.14% to −2.44%, p<0.001). DALY rates showed the steepest reduction of 59.5% from 37.59 (95% UI: 31.75–44.76) to 15.22 (95% UI: 10.67–17.57) per 100,000 (annual change: −3.18%, 95% UI: −3.51% to −2.84%,

**Data availability statement:** All data underlying the findings described in this manuscript are fully available without restriction from public repositories. The primary datasets analyzed during the current study are available from the Global Health Data Exchange (GHDx), Institute for Health Metrics and Evaluation, University of Washington (http://ghdx.healthdata.org/). Specifically, the Global Burden of Disease Study 2021 results for acute carbon monoxide poisoning, including age-standardized incidence, prevalence, mortality, and disability-adjusted life years (DALYs) data from 1990-2021 covering 204 countries and territories, are freely accessible after user registration. Socio-demographic Index (SDI) data by country and region are also available from the same source. Geographic base map data used for spatial visualization were obtained from Natural Earth (https://www.naturalearthdata.com/), which are in the public domain. All processed datasets and analysis code supporting the conclusions of this article are available from the corresponding author upon reasonable request.

**Funding:** The author(s) received no specific funding for this work.

**Competing interests:** The authors have declared that no competing interests exist.

**Abbreviations:** APC: Annual Percentage Change; AAPC: Average Annual Percentage Change; ARIMA: Autoregressive Integrated Moving Average; ASR: Age-Standardized Rate; CO: Carbon monoxide; DALY: Disability-Adjusted Life Year; GBD: Global Burden of Disease; GHDx: Global Health Data Exchange; LISA: Local Indicators of Spatial Association; SDI: Socio-Demographic Index; UI: Uncertainty Interval; WHO: World Health Organization; YLD: Years Lived with Disability; YLL: Years of Life Lost.

$p < 0.001$). Eastern Europe demonstrated the highest burden (37.98 per 100,000 in 2021). Males experienced significantly higher mortality than females (0.50 vs 0.20 per 100,000, $p < 0.001$). SDI analysis revealed an inverted U-shaped relationship (Spearman's $r = 0.76$, $p < 0.001$), with peak burden at moderate development levels (SDI: 0.6–0.7).

## Conclusions

These findings directly address previous research gaps by demonstrating: (1) faster mortality decline than incidence decline indicates improved global treatment capabilities; (2) the SDI-burden relationship identifies moderate-development countries as priority intervention targets; (3) significant male predominance (2.5-fold higher mortality) supports gender-specific prevention programs; and (4) persistent Eastern European hotspots require targeted infrastructure improvements. Predictive models forecast continued decline through 2050 and enable evidence-based healthcare planning. This comprehensive analysis provides the first multi-dimensional global assessment, offering crucial evidence for differentiated prevention strategies worldwide.

## Introduction

Carbon monoxide (CO) poisoning is one of the most common causes of fatal poisoning worldwide [1]. The burden of CO poisoning remains significant despite its preventable nature. Previous research has estimated that approximately 970,000 poisoning incidents occur annually worldwide, resulting in around 41,000 deaths [2]. CO is a colorless, odorless, and tasteless toxic gas produced by the incomplete combustion of carbon-containing fuels. Common sources include faulty heating systems, poorly ventilated cooking appliances, vehicle exhaust, and the burning of charcoal or other fuels in enclosed spaces [3].

Carbon monoxide exposure occurs through multiple pathways. Common sources include faulty heating systems, poorly ventilated cooking appliances, vehicle exhaust in enclosed spaces, and fuel-burning equipment such as generators [4]. Motor vehicle exhaust represents a significant source of CO exposure, particularly from stationary vehicles in enclosed spaces [5]. The health impacts range from acute symptoms including headache, dizziness, and nausea at concentrations of 50–100 ppm to severe neurological damage, cardiac arrhythmias, and death at concentrations exceeding 400 ppm [6]. Long-term sequelae among survivors include persistent neurological deficits, cognitive impairment, and increased risk of delayed neurological sequelae affecting 10–32% of patients [6].

The health impacts of CO poisoning can be severe and far-reaching. Acute exposure can lead to symptoms ranging from mild (headache, dizziness, nausea) to severe (loss of consciousness, heart damage, and death) [7]. Long-term sequelae among survivors may include neurological deficits, cognitive impairment, and cardiovascular complications. The economic burden is also substantial, encompassing direct medical costs, lost productivity, and long-term care expenses [8].

Several studies [9–11] have attempted to characterize the burden of CO poisoning at various levels. Long [9] et al. examined global mortality trends from 1990–2017, while Cui [10] et al. focused specifically on China's burden from 1990–2019. The Lancet Public Health [11] provided insights into mortality patterns from 2000–2021. However, existing research has several important limitations.

First, previous studies [9–11] primarily focused on mortality rates alone or were limited to specific regions like China. There is a lack of comprehensive analysis incorporating incidence, mortality, and disability-adjusted life years (DALYs) at the global level. This multi-dimensional understanding is crucial for fully grasping the public health impact of CO poisoning.

Second, although some studies [9–11] have examined regional variations in CO poisoning burden, they failed to systematically analyze the relationship between socio-demographic index (SDI) and CO poisoning burden at both regional and national levels. Understanding this relationship is crucial for policy-making and resource allocation, particularly in identifying vulnerable populations and regions.

Third, while existing studies [9–11] have described historical trends, there is no long-term prediction model for CO poisoning burden. Such predictions are essential for future healthcare planning and prevention strategies, especially given changing global patterns of energy use and climate.

Fourth, although gender disparities in CO poisoning have been noted in previous studies [9–11], systematic analysis of sex-specific patterns in incidence, mortality and DALYs across different regions and time periods is lacking. Understanding these patterns is vital for developing targeted prevention strategies.

Additionally, most existing studies [9–11] are descriptive in nature and do not provide in-depth analysis of geographical gradients or explore the complex factors contributing to regional variations in CO poisoning burden. Such detailed geographical analysis could reveal important patterns in disease burden distribution and help identify region-specific risk factors.

To address these significant gaps in knowledge, we conducted a comprehensive global analysis of CO poisoning burden using data from the Global Burden of Disease (GBD) Study 1990–2021. Our study aims were to: (1) quantify the global burden of CO poisoning across multiple dimensions including incidence, mortality, and disability-adjusted life years (DALYs); (2) analyze the relationship between socio-demographic development and CO poisoning burden at global, regional, and national levels; (3) develop predictive models estimating the future CO poisoning burden through 2050; (4) examine detailed geographical and sex-specific patterns in disease burden distribution; (5) explore the implications of these patterns for public health policy and prevention strategies.

This comprehensive analysis provides crucial insights for informing evidence-based policies and interventions to reduce CO poisoning burden worldwide. Understanding these patterns and trends is essential for developing targeted prevention strategies and allocating resources effectively to reduce the preventable burden of CO poisoning.

## Methods

### Study design

We conducted a comprehensive analysis of acute carbon monoxide poisoning using the Global Burden of Disease (GBD) 2021 database, accessed through the Global Health Data Exchange (GHDx) platform [12]. The analysis covered the period from 1990 to 2021, encompassing data from 204 countries and territories. Our research framework adhered to the GATHER guidelines for health estimates reporting [13]. The data were accessed for research purposes on March 15, 2021.

This study was conducted in accordance with the ethical standards of the institutional and/or national research committee and with the 1964 Helsinki Declaration and its later amendments or comparable ethical standards. The need for informed consent was waived by the ethics committee because the study involved the analysis of fully anonymized retrospective data from the Global Burden of Disease Study 2021. All data were accessed after complete anonymization, and no individual identifiers were included in the analysis. All analyses were conducted in compliance with ethical guidelines for human subjects research.

## Analytical framework

The geographical distribution of disease burden was evaluated using spatial statistics, incorporating both Global Moran's I for overall clustering assessment and LISA analysis for identifying regional patterns [14]. We examined the relationship between disease burden and development level by analyzing health metrics against the Socio-demographic Index (SDI), with nations grouped into SDI quintiles.

Temporal trend analysis employed joinpoint regression methodology, utilizing both APC and AAPC metrics [15]. The analysis parameters allowed for a maximum of three joinpoints, with optimal model selection guided by Monte Carlo permutation testing. Long-term projections extending to 2050 were generated using a combination of ARIMA modeling and exponential smoothing techniques [16], with model selection based on AIC values and validated through rolling-origin cross-validation. These three methods are complementary and form a comprehensive analytical framework: joinpoint regression describes historical trends, spatial statistics reveals geographic patterns, and ARIMA modeling predicts future trends. This multi-method integration provides a complete assessment of the global burden of CO poisoning. All methods were validated through appropriate statistical tests, model diagnostics, and uncertainty quantification. The combination approach enhances the robustness of our findings and provides multiple perspectives on the same epidemiological phenomenon.

## Burden metrics

We assessed disease burden through multiple indicators, primarily focusing on DALYs (calculated as the sum of YLLs and YLDs), along with incidence, prevalence, and mortality rates [17]. To ensure comparability across regions and time periods, all rates were age-standardized using the WHO standard population. Statistical uncertainty was addressed through 95% uncertainty intervals, with significance determined at $p < 0.05$.

## Computational tools

Analyses were performed using a combination of statistical software packages: R 4.1.0 and Stata 16 for primary analyses, the National Cancer Institute's Joinpoint Regression Program (version 4.9.1.0) for trend assessment, and ArcGIS Pro/ QGIS 3.16 for spatial visualization.

## Results

### 1. Epidemiological patterns and trends of acute carbon monoxide poisoning, 1990–2021

**1.1 Global burden and temporal trends (Table 1).** In 2021, there were 294,173.15 (95% UI: 209,197.11−395,043.48) cases of acute carbon monoxide poisoning globally, with an age-standardized incidence rate (ASR) of 7.87 (95% UI: 5.54–10.81) per 100,000 population. This represents a substantial reduction from 1990, when there were 693,139.82 (95% UI: 469,815.01−981,111.48) cases and an ASR of 12.13 (95% UI: 8.30–17.00) per 100,000 population. The estimated annual percentage change between 1990 and 2021 was −1.16% (95% UI: −1.35% to −0.96%).

Sex-specific patterns showed that in 1990, females had a higher ASR (12.92 per 100,000; 95% UI: 8.69–18.19) than males (11.35 per 100,000; 95% UI: 7.88–15.56). By 2021, both sexes showed decreased rates, with females at 8.16 (95% UI: 5.58–11.48) and males at 7.60 (95% UI: 5.46–10.26) per 100,000 population. The annual percentage change was −1.26% (95% UI: −1.44% to −1.07%) for females and −1.04% (95% UI: −1.24% to −0.84%) for males.

Across SDI quintiles, high-SDI regions recorded the highest ASR in 1990 at 24.77 (95% UI: 16.00–35.74) per 100,000, decreasing to 17.77 (95% UI: 11.48−25.96) in 2021. High-middle SDI regions showed similar patterns, with ASRs of 23.58 (95% UI: 16.66−31.97) in 1990 and 17.85 (95% UI: 12.82−24.28) in 2021. Low-SDI regions maintained the lowest rates throughout the study period, with ASRs of 3.39 (95% UI: 2.13−4.96) in 1990 and 2.78 (95% UI: 1.87−4.00) in 2021.

**Table 1. Global age-standardized incidence rates of acute carbon monoxide poisoning (1990-2021).**

| Characteristics | Incidence (95% uncertainty interval) | | | | |
|---|---|---|---|---|---|
| | Number of cases, 1990 | ASR per 100000 population, 1990 | Number of cases, 2021 | ASR per 100000 population, 2021 | Estimated annual percentage change, 1990–2021 |
| **Location** | | | | | |
| Global | 693139.82(469815.01,981111.48) | 12.13(8.30,17.00) | 294173.15(209197.11,395043.48) | 7.87(5.54,10.81) | −1.16(−1.35,-0.96) |
| **SEX** | | | | | |
| Male | 328711.45(224639.10,458237.94) | 11.35(7.88,15.56) | 294173.15(209197.11,395043.48) | 7.60(5.46,10.26) | −1.04(−1.24,-0.84) |
| Female | 364428.36(242796.67,517300.42) | 12.92(8.69,18.19) | 8306466.88(211120.03,426664.43) | 8.16(5.58,11.48) | −1.26(−1.44,-1.07) |
| **SDI** | | | | | |
| High | 198935.21(129323.21,286238.29) | 24.77(16.00,35.74) | 151650.80(102859.16,211801.98) | 17.77(11.48,25.96) | −0.50(−0.81,-0.19) |
| High middle | 256050.68(179270.77,349599.88) | 23.58(16.66,31.97) | 196799.02(143406.36,255270.12) | 17.85(12.82,24.28) | −0.84(−1.08,-0.60) |
| Middle | 154919.60(101287.58,225975.71) | 7.86(5.21,11.28) | 148982.45(104247.71,206861.25) | 6.59(4.60,9.21) | −0.31(−0.52,-0.10) |
| Low middle | 58389.08(34906.93,88761.71) | 3.96(2.42,5.90) | 63033.75(39297.60,94359.63) | 3.08(1.96,4.56) | −0.75(−0.93,-0.57) |
| Low | 23855.82(14849.28,35667.90) | 3.39(2.13,4.96) | 39546.60(25813.05,57782.10) | 2.78(1.87,4.00) | −0.61(−0.73,-0.48) |
| **GBD region** | | | | | |
| Andean Latin America | 17726.94(12142.04,25222.30) | 5.20(2.99,8.15) | 17856.25(12318.94,24992.04) | 5.24(3.14,8.21) | 0.23(0.09,0.38) |
| Australasia | 19023.75(10938.11,30400.34) | 14.57(9.19,21.82) | 18737.08(11288.82,29238.40) | 12.53(8.10,18.50) | −0.46(−0.58,-0.35) |
| Caribbean | 45513.59(30788.93,65582.50) | 10.07(6.38,15.16) | 22885.35(15479.23,32773.32) | 9.95(6.27,14.98) | 0.10(−0.10,0.30) |
| Central Asia | 146853.21(98903.12,212419.73) | 22.89(15.90,32.12) | 156104.38(114589.18,204066.93) | 18.63(12.78,26.62) | −0.53(−0.70,-0.36) |
| Central Europe | 233.18(135.21,365.42) | 39.11(26.39,56.82) | 473.67(292.14,713.62) | 27.93(17.88,41.60) | −0.88(−1.12,-0.65) |
| Central Latin America | 71937.76(46324.29,103772.32) | 13.51(7.64,21.48) | 55007.05(37113.48,76629.75) | 9.76(5.79,15.24) | −0.89(−1.18,-0.60) |
| Central sub Saharan Africa | 120813.46(88810.99,155630.71) | 2.72(1.77,3.90) | 67288.99(51320.35,84904.71) | 2.31(1.61,3.29) | −0.54(−0.61,-0.47) |
| East Asia | 2812.58(1789.11,4158.10) | 11.17(7.70,15.68) | 3239.31(2150.70,4572.90) | 12.87(9.26,17.06) | 0.97(0.71,1.24) |
| Eastern Europe | 50421.41(33402.82,72531.56) | 55.71(41.19,72.97) | 22839.71(15164.33,32314.86) | 37.98(28.51,49.23) | −1.65(−2.08,-1.22) |
| Eastern sub Saharan Africa | 12807.52(7926.84,19414.28) | 3.59(2.36,5.24) | 15865.56(10592.01,22946.20) | 2.76(1.92,3.87) | −0.88(−0.98,-0.77) |
| High-income Asia Pacifc | 65618.56(40399.00,96858.55) | 32.77(21.77,46.85) | 64048.97(42794.83,90451.93) | 19.63(12.42,29.50) | −1.61(−1.75,-1.47) |
| High-income North America | 3981.36(2506.79,6053.51) | 24.71(15.13,37.26) | 4327.88(2744.06,6476.59) | 20.23(13.00,29.09) | 0.64(−0.16,1.44) |
| North Africa and Middle East | 2487.97(1394.06,3978.93) | 6.82(4.46,9.79) | 3473.55(2091.08,5451.20) | 6.27(4.20,8.89) | −0.28(−0.45,-0.11) |
| Oceania | 9773.97(3624.09,18548.81) | 2.90(1.72,4.42) | 5684.71(3070.07,9373.61) | 2.98(1.85,4.43) | 0.24(0.05,0.44) |
| South Asia | 28028.39(15540.38,45387.22) | 2.86(1.66,4.36) | 23333.69(13951.02,36344.19) | 1.98(1.18,3.01) | −1.16(−1.40,-0.92) |
| Southeast Asia | 2092.18(1338.52,3038.87) | 3.39(2.00,5.35) | 3851.93(2531.49,5752.41) | 2.84(1.71,4.45) | −0.41(−0.67,-0.15) |
| Southern Latin America | 29249.23(18928.36,42828.76) | 24.92(15.47,37.50) | 40452.74(26969.25,57497.74) | 26.73(17.66,38.98) | 0.46(0.19,0.72) |
| Southern Sub-Saharan Africa | 40203.90(22969.17,62557.45) | 4.61(3.09,6.60) | 37456.27(21965.29,57577.96) | 3.04(2.13,4.28) | −1.41(−1.48,-1.34) |
| Tropical Latin America | 9474.21(6130.21,14074.91) | 5.60(2.21,10.38) | 13963.65(9202.91,20591.85) | 2.71(1.42,4.52) | −2.05(−2.24,-1.86) |
| Western Europe | 10891.55(6953.62,16251.54) | 22.04(13.87,32.09) | 21212.82(13837.67,30711.96) | 17.08(11.03,24.98) | −0.60(−0.73,-0.46) |
| Western Sub-Saharan Africa | 3195.10(2105.60,4624.59) | 3.89(2.49,5.73) | 2536.50(1757.36,3581.84) | 3.15(2.09,4.52) | −0.64(−0.83,-0.46) |

In the GBD regional analysis, Eastern Europe showed the highest ASR in 1990 (55.71 per 100,000; 95% UI: 41.19–72.97), followed by High-income Asia Pacific (32.77 per 100,000; 95% UI: 21.77–46.85). By 2021, Eastern Europe's ASR had decreased to 37.98 (95% UI: 28.51–49.23). East Asia showed an increase in ASR from 11.17 (95% UI: 7.70–15.68) to 12.87 (95% UI: 9.26–17.06), with an annual percentage change of 0.97% (95% UI: 0.71–1.24%). Southern Latin America also demonstrated an increase, with an annual percentage change of 0.46% (95% UI: 0.19–0.72%).

**1.2  Deaths and mortality trends (Table 2).**  Global deaths from acute carbon monoxide poisoning decreased from 36,816.05 (95% UI: 31,520.91–44,285.07) in 1990–28,946.78 (95% UI: 19,894.89–33,510.18) in 2021. The age-standardized death rate (ASR) showed a more pronounced decline, from 0.76 (95% UI: 0.66–0.91) to 0.35 (95% UI: 0.24–0.40) per 100,000 population, with an estimated annual percentage change of −2.79% (95% UI: −3.14% to −2.44%).

Sex-stratified analysis revealed consistently higher mortality rates among males. In 1990, the male ASR was 1.07 (95% UI: 0.88–1.39) compared to 0.46 (95% UI: 0.32–0.63) for females per 100,000 population. By 2021, these rates had decreased to 0.50 (95% UI: 0.36–0.63) for males and 0.20 (95% UI: 0.10–0.25) for females, with similar annual percentage changes of −2.756% (95% UI: −3.12% to −2.39%) and −2.771% (95% UI: −3.09% to −2.46%), respectively.

Among SDI quintiles, high-middle SDI regions recorded the highest ASR in 1990 at 1.79 (95% UI: 1.66–2.05) per 100,000, decreasing to 0.76 (95% UI: 0.59–0.86) in 2021. Low-middle SDI regions maintained the lowest rates, declining from 0.20 (95% UI: 0.11–0.27) to 0.12 (95% UI: 0.07–0.16) per 100,000 population.

Geographically, Eastern Europe showed the highest mortality burden, with an ASR of 4.44 (95% UI: 4.33–4.52) per 100,000 in 1990, decreasing to 2.10 (95% UI: 1.95–2.25) in 2021. Central Europe experienced the steepest decline with an annual percentage change of −5.15% (95% UI: −5.42% to −4.89%). Notably, some regions showed increasing trends, including High-income North America (1.16%; 95% UI: 0.88–1.45%) and Andean Latin America (2.42%; 95% UI: 1.83–3.02%). East Asia maintained a substantial burden throughout the study period, with death numbers of 14,111.63 (95% UI: 10,243.22−21,401.62) in 1990 and 13,588.38 (95% UI: 6,818.19−17,624.96) in 2021.

**1.3  Disability-adjusted life years (Table 3).**  The global burden of acute carbon monoxide poisoning measured in disability-adjusted life years (DALYs) decreased substantially from 1,991,539.89 (95% UI: 1,664,186.66−2,374,749.90) in 1990–1,228,861.20 (95% UI: 867,811.13−1,414,898.57) in 2021. The age-standardized DALY rate showed a marked reduction from 37.59 (95% UI: 31.75–44.76) to 15.22 (95% UI: 10.67–17.57) per 100,000 population, with an estimated annual percentage change of −3.18% (95% UI: −3.51% to −2.84%).

The sex-specific burden demonstrated pronounced disparities. Males experienced higher DALY rates, with an ASR of 51.84 (95% UI: 41.59–64.96) per 100,000 in 1990, decreasing to 21.52 (95% UI: 15.32–26.87) in 2021. Female rates declined from 23.13 (95% UI: 15.24–31.26) to 8.94 (95% UI: 4.58–10.63) per 100,000, with annual percentage changes of −3.13% (95% UI: −3.49% to −2.77%) for males and −3.23% (95% UI: −3.53% to −2.93%) for females.

Across SDI quintiles, high-middle SDI regions showed the highest burden with an ASR of 87.55 (95% UI: 80.24–100.93) per 100,000 in 1990, declining to 33.83 (95% UI: 26.92–38.06) in 2021. Low-middle SDI regions maintained the lowest rates, decreasing from 10.69 (95% UI: 5.41–15.27) to 6.23 (95% UI: 3.53–8.53) per 100,000 population.

In the regional analysis, Eastern Europe recorded the highest ASR in 1990 at 212.93 (95% UI: 207.51–217.50) per 100,000, dropping to 95.02 (95% UI: 88.66–101.06) in 2021. Central Europe showed the steepest decline with an annual percentage change of −5.40% (95% UI: −5.67% to −5.14%). Conversely, several regions exhibited increasing trends, including Andean Latin America (1.91%; 95% UI: 1.35–2.48%) and High-income North America (1.02%; 95% UI: 0.70–1.33%). East Asia demonstrated a substantial reduction in ASR from 67.98 (95% UI: 49.48–100.37) to 34.61 (95% UI: 17.91–43.44) per 100,000 population.

**Global geographic distribution**

**2.1  Geographic distribution of age-standardized incidence rates.**  The age-standardized incidence rates (ASR) of acute carbon monoxide poisoning showed marked geographic variation across the global regions (Fig 1). The highest

**Table 2. Global age-standardized mortality rates of acute carbon monoxide poisoning (1990-2021).**

| Characteristics | Deaths (95% uncertainty interval) | | | | |
|---|---|---|---|---|---|
| | Number of cases, 1990 | ASR per 100000 population, 1990 | Number of cases, 2021 | ASR per 100000 population, 2021 | Estimated annual percentage change, 1990–2021 |
| **Location** | | | | | |
| Global | 36816.05(31520.91,44285.07) | 0.76(0.66,0.91) | 28946.78(19894.89,33510.18) | 0.35(0.24,0.40) | −2.79(−3.14,-2.44) |
| **SEX** | | | | | |
| Male | 25574.21(20701.72,32318.68) | 1.07(0.88,1.39) | 20168.71(14337.62,25421.30) | 0.50(0.36,0.63) | −2.756(−3.12,-2.39) |
| Female | 11241.85(7609.71,15551.99) | 0.46(0.32,0.63) | 8778.07(4290.75,10787.13) | 0.20(0.10,0.25) | −2.771(−3.09,-2.46) |
| **SDI** | | | | | |
| High | 1054.73(439.33,1779.22) | 0.42(0.39,0.44) | 1165.51(580.18,1533.12) | 0.22(0.20,0.23) | −1.85(−2.07,-1.62) |
| High middle | 508.27(489.00,528.85) | 1.79(1.66,2.05) | 395.61(358.88,441.29) | 0.76(0.59,0.86) | −3.37(−3.93,-2.80) |
| Middle | 794.06(507.64,963.83) | 0.73(0.54,1.03) | 313.24(282.27,404.35) | 0.40(0.21,0.49) | −1.75(−2.00,-1.50) |
| Low middle | 1274.36(1162.78,1361.64) | 0.20(0.11,0.27) | 927.56(818.72,1070.26) | 0.12(0.07,0.16) | −1.89(−2.03,-1.74) |
| Low | 136.15(40.82,193.44) | 0.29(0.16,0.45) | 140.26(68.54,238.92) | 0.18(0.12,0.32) | −1.62(−1.71,-1.54) |
| **GBD region** | | | | | |
| Andean Latin America | 11.13(5.36,16.71) | 0.03(0.02,0.05) | 45.58(30.48,57.32) | 0.07(0.05,0.09) | 2.42(1.83,3.02) |
| Australasia | 23.82(22.60,24.97) | 0.11(0.11,0.12) | 28.17(26.67,29.72) | 0.08(0.08,0.09) | −1.70(−2.48,-0.92) |
| Caribbean | 73.63(53.89,104.69) | 0.22(0.16,0.29) | 44.33(29.46,59.88) | 0.09(0.06,0.13) | −2.52(−2.96,-2.08) |
| Central Asia | 1274.36(1162.78,1361.64) | 2.06(1.90,2.19) | 927.56(818.72,1070.26) | 0.98(0.86,1.12) | −3.79(−4.50,-3.07) |
| Central Europe | 2041.25(1912.23,2153.07) | 1.58(1.48,1.67) | 583.63(538.76,631.37) | 0.35(0.33,0.38) | −5.15(−5.42,-4.89) |
| Central Latin America | 508.27(489.00,528.85) | 0.34(0.33,0.35) | 395.61(358.88,441.29) | 0.16(0.14,0.17) | −2.61(−2.82,-2.39) |
| Central sub Saharan Africa | 68.41(34.12,166.40) | 0.18(0.09,0.44) | 77.66(15.49,386.59) | 0.09(0.02,0.45) | −2.12(−2.47,-1.76) |
| East Asia | 14111.63(10243.22,21401.62) | 1.30(0.94,1.97) | 13588.38(6818.19,17624.96) | 0.79(0.40,1.01) | −1.36(−1.68,-1.04) |
| Eastern Europe | 10975.80(10709.96,11176.15) | 4.44(4.33,4.52) | 5642.24(5227.16,6079.50) | 2.10(1.95,2.25) | −3.42(−4.31,-2.53) |
| Eastern sub Saharan Africa | 268.17(140.25,690.06) | 0.21(0.10,0.52) | 208.84(54.06,978.01) | 0.08(0.02,0.37) | −3.06(−3.36,-2.76) |
| High-income Asia Pacifc | 794.06(507.64,963.83) | 0.44(0.27,0.53) | 313.24(282.27,404.35) | 0.12(0.11,0.16) | −3.49(−3.97,-3.02) |
| High-income North America | 822.71(800.43,837.81) | 0.27(0.26,0.28) | 1303.77(1255.79,1345.75) | 0.31(0.30,0.32) | 1.16(0.88,1.45) |
| North Africa and Middle East | 2471.39(1433.07,3282.09) | 0.82(0.50,1.07) | 2770.75(1413.49,3626.76) | 0.48(0.25,0.62) | −1.63(−1.73,-1.53) |
| Oceania | 14.76(5.98,24.62) | 0.26(0.11,0.46) | 24.81(10.87,44.74) | 0.20(0.09,0.36) | −0.98(−1.03,-0.94) |
| South Asia | 1054.73(439.33,1779.22) | 0.11(0.05,0.17) | 1165.51(580.18,1533.12) | 0.07(0.03,0.09) | −1.49(−1.59,-1.38) |
| Southeast Asia | 578.41(181.28,821.89) | 0.15(0.04,0.21) | 712.22(275.08,974.81) | 0.10(0.04,0.14) | −1.25(−1.30,-1.21) |
| Southern Latin America | 121.45(103.39,131.14) | 0.25(0.21,0.27) | 245.89(232.87,258.93) | 0.33(0.31,0.35) | 0.77(0.04,1.50) |
| Southern Sub-Saharan Africa | 136.15(40.82,193.44) | 0.25(0.08,0.36) | 140.26(68.54,238.92) | 0.17(0.08,0.29) | −1.37(−1.52,-1.22) |
| Tropical Latin America | 81.19(77.22,85.26) | 0.06(0.06,0.06) | 68.41(65.13,71.53) | 0.03(0.03,0.03) | −2.24(−2.66,-1.82) |
| Western Europe | 1262.70(1210.56,1295.79) | 0.28(0.27,0.29) | 536.83(500.25,560.65) | 0.09(0.08,0.09) | −3.59(−3.94,-3.24) |
| Western Sub-Saharan Africa | 122.05(52.76,204.83) | 0.12(0.05,0.19) | 123.09(73.44,368.12) | 0.05(0.03,0.14) | −2.93(−3.22,-2.65) |

**Table 3. Global age-standardized DALY rates of acute carbon monoxide poisoning (1990-2021).**

| Characteristics | DALYs (95% uncertainty interval) | | | | |
|---|---|---|---|---|---|
| | Number of cases, 1990 | ASR per 100000 population, 1990 | Number of cases, 2021 | ASR per 100000 population, 2021 | Estimated annual percentage change, 1990–2021 |
| **Location** | | | | | |
| Global | 1991539.89(1664186.66,2374749.90) | 37.59(31.75,44.76) | 1228861.20(867811.13,1414898.57) | 15.22(10.67,17.57) | −3.18(−3.51,-2.84) |
| **SEX** | | | | | |
| Male | 1371568.09(1084724.59,1717161.87) | 51.84(41.59,64.96) | 869791.56(622356.12,1085567.94) | 21.52(15.32,26.87) | −3.13(−3.49,-2.77) |
| Female | 619971.79(404516.50,837833.47) | 23.13(15.24,31.26) | 359069.63(185060.10,427811.11) | 8.94(4.58,10.63) | −3.23(−3.53,-2.93) |
| **SDI** | | | | | |
| High | 204572.05(189656.73,216175.29) | 22.79(21.03,24.10) | 131482.24(121542.74,141061.18) | 11.40(10.56,12.19) | −1.89(−2.12,-1.67) |
| High middle | 917444.00(842240.63,1063693.98) | 87.55(80.24,100.93) | 479786.22(383939.91,542794.21) | 33.83(26.92,38.06) | −3.62(−4.17,-3.06) |
| Middle | 670765.64(495400.21,912876.80) | 37.82(27.78,51.51) | 414933.91(228494.85,511805.42) | 16.70(9.17,20.34) | −2.53(−2.77,-2.29) |
| Low middle | 130396.47(63231.17,193448.73) | 10.69(5.41,15.27) | 120411.32(67754.07,164817.72) | 6.23(3.53,8.53) | −2.02(−2.15,-1.89) |
| Low | 66398.32(34227.76,110234.62) | 13.07(7.31,20.32) | 81550.42(53040.20,132928.44) | 7.89(5.21,13.35) | −1.69(−1.74,-1.63) |
| **GBD region** | | | | | |
| Andean Latin America | 820630.90(598757.83,1217182.22) | 2.17(1.32,3.02) | 511718.87(265448.76,653682.47) | 3.83(2.67,4.80) | 1.91(1.35,2.48) |
| Australasia | 5377.19(5078.92,5788.58) | 7.30(6.85,7.86) | 3630.56(3436.15,3861.47) | 5.05(4.71,5.46) | −1.80(−2.49,-1.11) |
| Caribbean | 44326.87(42618.63,46385.80) | 13.24(9.70,19.61) | 60909.71(58475.04,63353.25) | 6.40(4.17,8.82) | −2.12(−2.53,-1.70) |
| Central Asia | 96370.67(90201.29,102554.55) | 104.85(95.85,112.26) | 21108.45(19478.90,22867.45) | 46.49(41.04,54.26) | −3.97(−4.66,-3.28) |
| Central Europe | 70670.56(30531.07,124491.55) | 78.78(73.81,84.02) | 66611.15(36220.66,88601.96) | 16.08(14.79,17.47) | −5.40(−5.67,-5.14) |
| Central Latin America | 34257.43(32624.17,36216.62) | 19.28(18.46,20.23) | 20849.21(19029.68,23084.47) | 8.14(7.40,9.05) | −2.76(−2.95,-2.57) |
| Central sub Saharan Africa | 33379.17(11953.46,47448.27) | 7.63(4.13,18.03) | 37231.89(15737.43,51304.60) | 3.83(0.98,17.87) | −2.29(−2.60,-1.97) |
| East Asia | 46356.65(31026.47,55143.99) | 67.98(49.48,100.37) | 14237.66(12593.92,17902.02) | 34.61(17.91,43.44) | −2.10(−2.42,-1.78) |
| Eastern Europe | 71323.25(64693.73,77010.81) | 212.93(207.51,217.50) | 45707.49(40263.33,53471.48) | 95.02(88.66,101.06) | −3.56(−4.44,-2.67) |
| Eastern sub Saharan Africa | 158136.89(88173.65,218144.28) | 8.65(4.65,21.42) | 147819.73(77885.88,199152.04) | 3.40(1.13,14.75) | −3.11(−3.42,-2.80) |
| High-income Asia Pacifc | 497098.53(484659.98,507418.72) | 26.55(17.34,31.73) | 222147.26(206162.01,237843.44) | 7.15(6.30,9.15) | −3.72(−4.14,-3.29) |
| High-income North America | 922.24(359.49,1595.04) | 15.23(14.65,15.93) | 1547.51(698.44,2694.01) | 16.20(15.58,16.84) | 1.02(0.70,1.33) |
| North Africa and Middle East | 4872.01(3490.01,7457.12) | 43.61(25.48,57.75) | 2887.44(1938.23,3912.98) | 23.53(12.42,31.64) | −1.92(−2.04,-1.80) |
| Oceania | 16798.08(9169.44,38581.06) | 13.41(5.53,22.21) | 11857.83(4001.58,51031.32) | 10.56(4.68,18.57) | −0.77(−0.82,-0.72) |
| South Asia | 6033.15(3261.37,9495.43) | 6.04(2.71,10.04) | 7194.91(4794.19,17292.55) | 3.54(1.90,4.70) | −1.66(−1.81,-1.50) |
| Southeast Asia | 7581.29(6452.38,8359.07) | 7.37(2.62,10.42) | 11986.36(11283.92,12755.88) | 5.08(2.18,6.98) | −1.22(−1.27,-1.17) |
| Southern Latin America | 4310.08(1893.94,10214.44) | 15.18(12.93,16.74) | 4247.92(1122.04,18970.74) | 17.91(16.80,19.09) | 0.42(−0.26,1.10) |
| Southern Sub-Saharan Africa | 1520.86(1426.32,1636.44) | 16.45(5.55,23.04) | 1547.84(1438.41,1673.98) | 11.24(5.62,18.63) | −1.34(−1.53,-1.14) |
| Tropical Latin America | 9749.06(3217.44,13696.04) | 3.51(3.32,3.76) | 9405.91(4677.61,15655.28) | 1.54(1.45,1.64) | −2.44(−2.80,-2.07) |
| Western Europe | 60940.55(57943.42,64260.40) | 15.45(14.70,16.26) | 23645.84(21516.11,26366.24) | 4.99(4.53,5.56) | −3.36(−3.67,-3.05) |
| Western Sub-Saharan Africa | 884.46(521.45,1289.67) | 4.60(2.36,7.34) | 2567.65(1796.32,3211.73) | 2.11(1.38,5.41) | −2.59(−2.79,-2.38) |

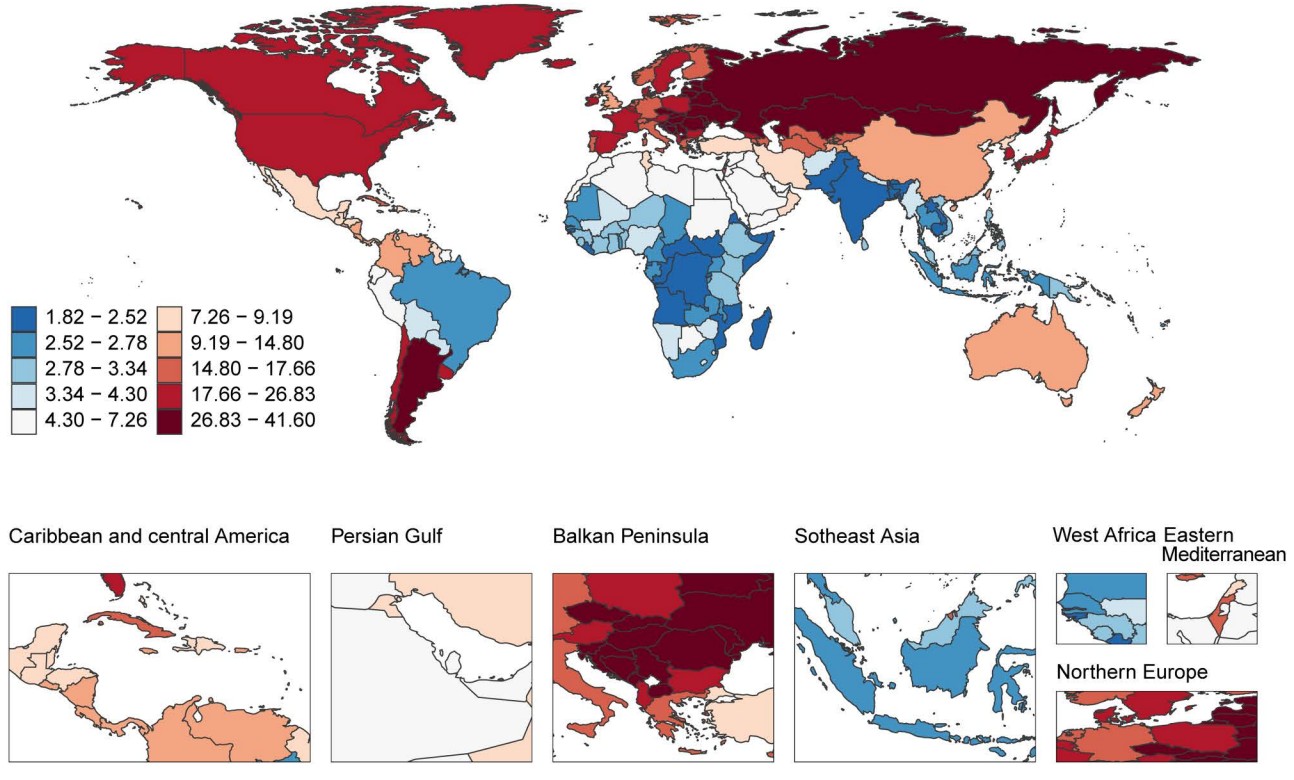

**Fig 1. Geographic distribution of age-standardized incidence rates of acute carbon monoxide poisoning (per 100,000 population) in 2021.**
Disease burden data from Global Burden of Disease Study 2021; Map data from Natural Earth (https://www.naturalearthdata.com/), public domain.

rates were observed in the Balkan Peninsula and Northern Europe, with ASRs ranging from 26.83 to 41.60 per 100,000 population. The second-highest burden was found in North America and parts of Northern Asia, where ASRs ranged from 17.66 to 26.83 per 100,000 population.

Australia and parts of Central America displayed moderate incidence rates, falling within the range of 9.19 to 14.80 per 100,000 population. The Persian Gulf region and parts of Central Asia showed rates between 4.30 and 7.26 per 100,000 population.

Lower incidence rates were predominantly observed in Africa and Southeast Asia, where most regions recorded ASRs between 1.82 and 3.34 per 100,000 population. Specifically, Southeast Asian countries demonstrated rates in the lowest bracket (1.82–2.52 per 100,000 population), while most of sub-Saharan Africa showed rates between 2.52 and 3.34 per 100,000 population.

A clear gradient was visible from north to south in the Americas, with North America showing substantially higher rates (17.66–26.83 per 100,000) compared to South America, where rates varied but generally remained between 3.34 and 4.30 per 100,000 population, except for the southernmost regions which displayed higher rates.

**2.2 Geographic distribution of age-standardized mortality rates.** The age-standardized mortality rates (ASR) for acute carbon monoxide poisoning exhibited distinct geographical patterns across the world (Fig 2). The highest mortality rates were concentrated in Northern Asia and parts of Eastern Europe, where ASRs reached between 0.63 and 2.72 per 100,000 population. The Balkan Peninsula region also demonstrated notably high mortality rates, falling within the range of 0.32 to 0.63 per 100,000 population.

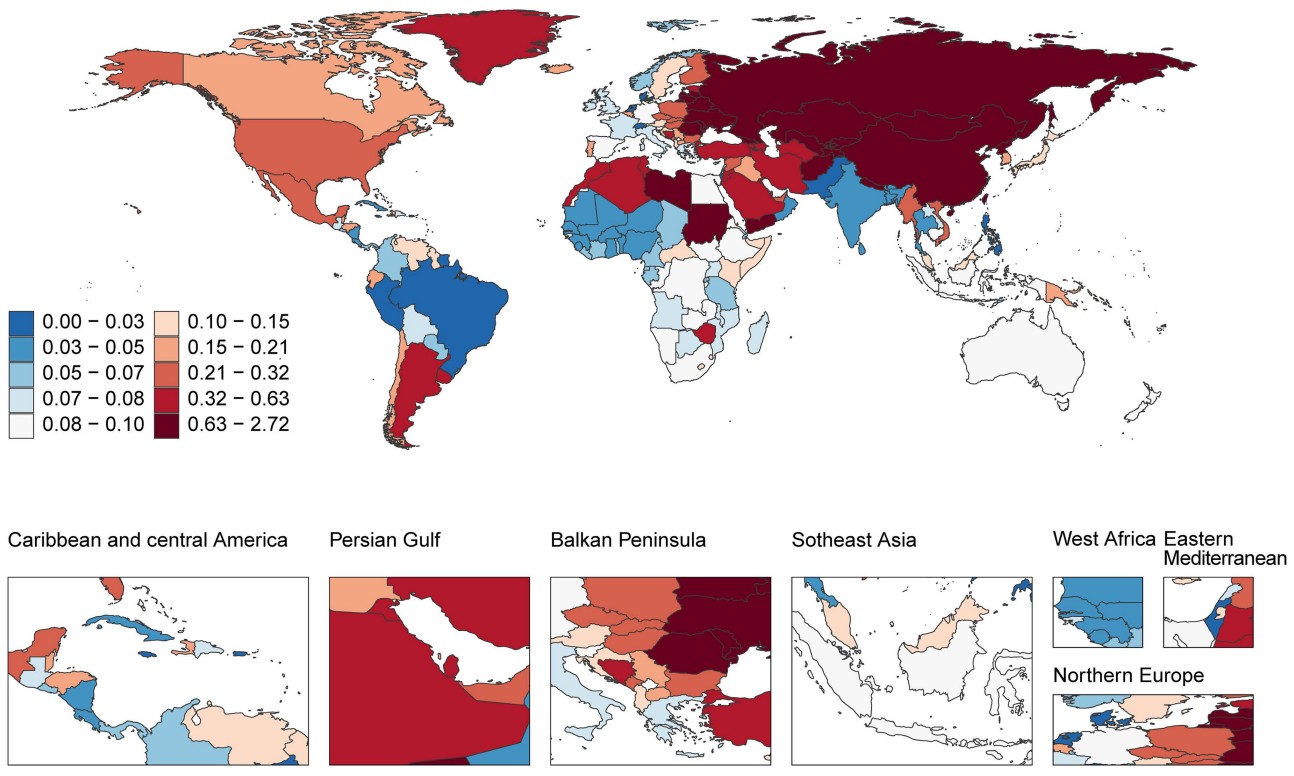

**Fig 2. Geographic distribution of age-standardized mortality rates of acute carbon monoxide poisoning (per 100,000 population) in 2021.** Disease burden data from Global Burden of Disease Study 2021; Map data from Natural Earth (https://www.naturalearthdata.com/), public domain.

The Persian Gulf region displayed elevated mortality rates, with ASRs ranging from 0.21 to 0.32 per 100,000 population. North America showed moderate mortality rates, with ASRs between 0.15 and 0.21 per 100,000 population. Similar levels were observed in parts of Central Asia and Northern Europe.

Lower mortality rates were documented across most of Southeast Asia and Africa, where ASRs generally ranged from 0.03 to 0.07 per 100,000 population. The lowest mortality rates (0.00–0.03 per 100,000 population) were observed in parts of South America, particularly in Brazil and surrounding regions.

A notable gradient was apparent in the Americas, with higher rates in North America (0.15–0.21 per 100,000) contrasting with lower rates in central regions, though Southern Latin America showed elevated rates comparable to those in North America. The mortality pattern in Europe displayed considerable heterogeneity, with a clear east-to-west gradient showing higher rates in Eastern Europe and lower rates in Western European countries.

**2.3 Geographic distribution of age-standardized DALY rates.** The age-standardized DALY rates for acute carbon monoxide poisoning showed substantial geographic variation worldwide (Fig 3). Northern Asia demonstrated the highest burden, with rates ranging from 30.55 to 120.20 DALYs per 100,000 population. The Balkan Peninsula and parts of Eastern Europe also exhibited notably high rates, falling within the range of 16.75 to 30.55 DALYs per 100,000 population.

The Persian Gulf region and North America showed moderately high DALY rates, ranging from 10.58 to 16.75 per 100,000 population. Central Asian countries displayed intermediate burden levels, with rates typically between 8.09 and 10.58 DALYs per 100,000 population.

Lower DALY rates were observed across most of Africa and Southeast Asia, where rates generally ranged from 2.53 to 4.30 per 100,000 population. The lowest burden was documented in parts of South America, particularly Brazil, with rates between 0.87 and 2.05 DALYs per 100,000 population.

The Americas showed a clear north-south divide in DALY rates, with North America experiencing higher rates (10.58–16.75 per 100,000) compared to Central America (5.46–8.09 per 100,000), though Southern Latin America demonstrated elevated rates similar to North American levels. European DALY rates exhibited marked regional variation, with a pronounced east-to-west gradient showing substantially higher rates in Eastern Europe compared to Western European nations.

## 3 Global analysis of the relationship between Socio-Demographic Index (SDI) and carbon monoxide poisoning disease burden

### 3.1 Regional-level analysis

**3.1.1 Regional association between SDI and age-standardized incidence rates.** The age-standardized incidence rates (ASR) of acute carbon monoxide poisoning showed a strong positive correlation with socio-demographic index (SDI) across 21 global regions (Spearman's r = 0.7611, p = 2.316e-142) (Fig 4). The relationship between SDI and incidence rates demonstrated distinct patterns across different regions.

Eastern Europe exhibited the highest incidence rates, reaching approximately 70 per 100,000 population at SDI levels around 0.6–0.7. Central Asia and High-income Asia Pacific regions showed the second-highest peaks, with incidence rates of about 30–40 per 100,000 population at similar SDI levels.

Western Europe and High-income North America displayed moderate incidence rates (20−30 per 100,000 population) at high SDI values (0.8–0.9). Australasia and Southern Latin America followed a similar pattern but with slightly lower rates.

Regions with lower SDI values (0.3–0.5), including South Asia, Southeast Asia, and Sub-Saharan African regions, consistently showed lower incidence rates, generally below 10 per 100,000 population. The global average curve demonstrated an overall increasing trend with SDI, peaking at moderate SDI levels (0.6–0.7) before showing a slight decline at the highest SDI values.

East Asia and Central Europe showed distinct trajectories, with rates increasing substantially as SDI increased from 0.4 to 0.7, followed by a plateau or slight decrease at higher SDI levels. The relationship between SDI and incidence rates appeared most pronounced in the mid-range of SDI values (0.5–0.7), where the greatest variation in rates was observed across regions.

**3.1.2 Regional association between SDI and age-standardized mortality rates.** The relationship between socio-demographic index (SDI) and age-standardized death rates (ASR) for acute carbon monoxide poisoning across 21 global regions showed a moderate positive correlation (Spearman's r = 0.3075, p = 7.627e-18) (Fig 5). The pattern of mortality rates demonstrated distinct regional variations across the SDI spectrum.

Eastern Europe exhibited the highest mortality rates, reaching approximately 7 deaths per 100,000 population at SDI values between 0.6 and 0.7. Central Asia showed the second highest peak, with mortality rates around 3 deaths per 100,000 population at similar SDI levels.

Most regions maintained relatively low mortality rates (below 1 per 100,000 population) across the SDI spectrum. High-income regions, including Western Europe, High-income North America, and Australasia, showed consistently low mortality rates despite their high SDI values (0.8–0.9).

The global average curve demonstrated a subtle increase with rising SDI values up to approximately 0.6, followed by a gradual decline at higher SDI levels. East Asia showed a distinctive pattern with elevated mortality rates at moderate SDI levels (0.6–0.7), while maintaining lower rates than Eastern Europe.

Regions with lower SDI values (0.3–0.5), including South Asia, Southeast Asia, and Sub-Saharan African regions, generally maintained low mortality rates below 0.5 per 100,000 population, showing minimal variation across different SDI values.

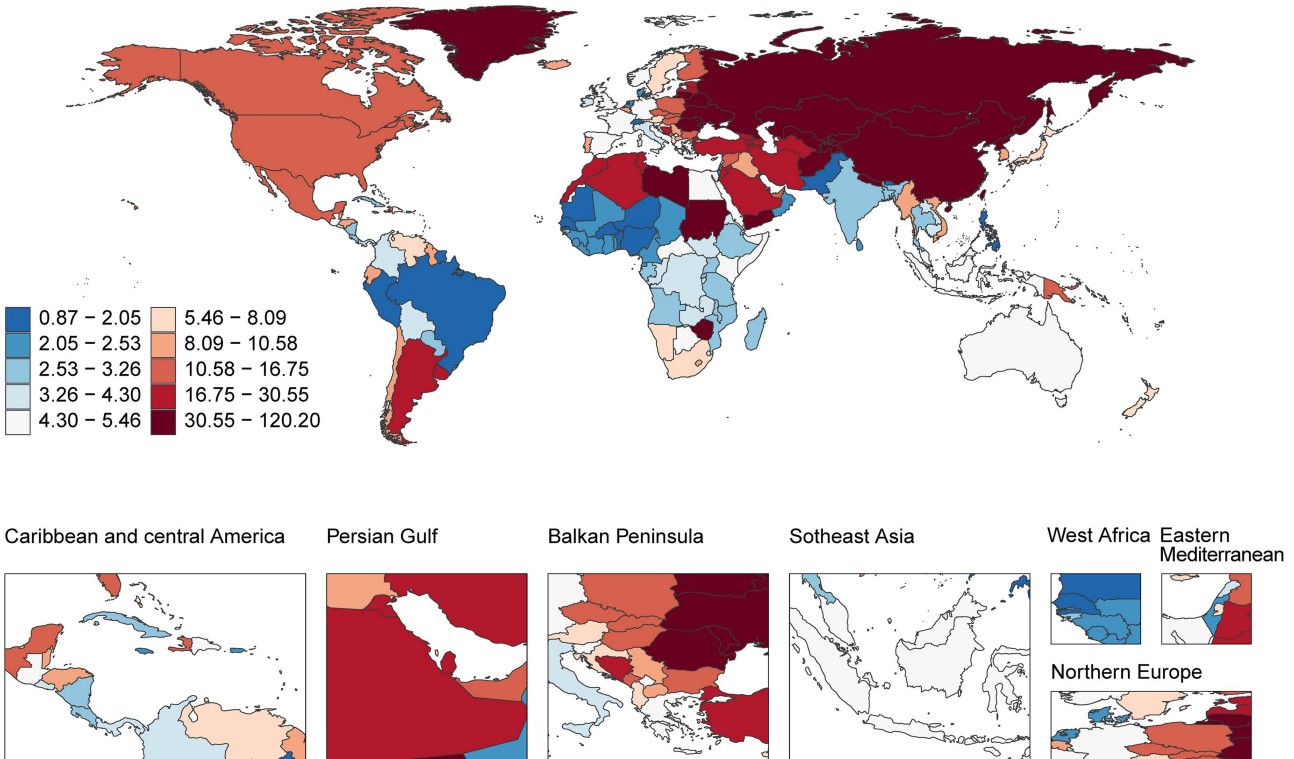

**Fig 3. Geographic distribution of age-standardized DALY rates of acute carbon monoxide poisoning (per 100,000 population) in 2021.** Disease burden data from Global Burden of Disease Study 2021; Map data from Natural Earth (https://www.naturalearthdata.com/), public domain.

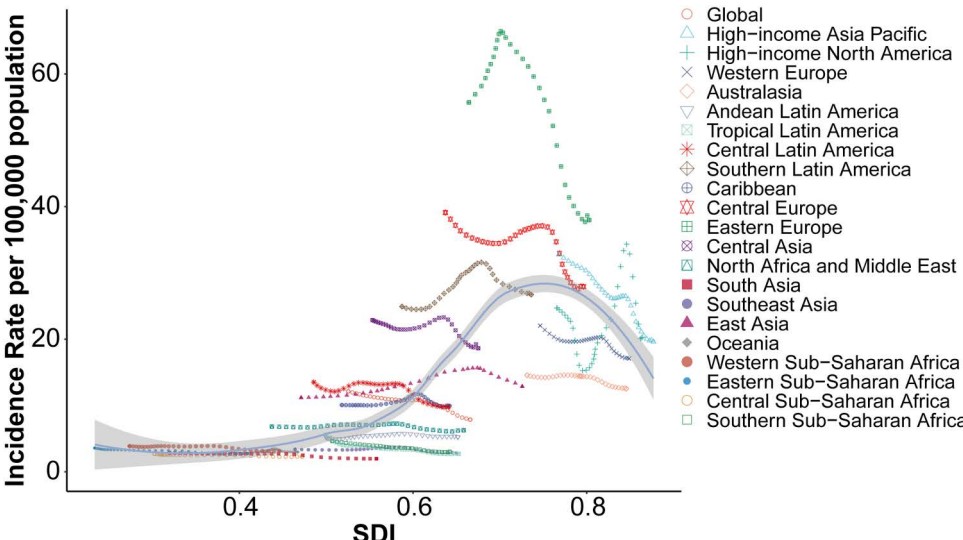

**Fig 4. Association between socio-demographic index and age-standardized incidence rates across 21 global regions.**

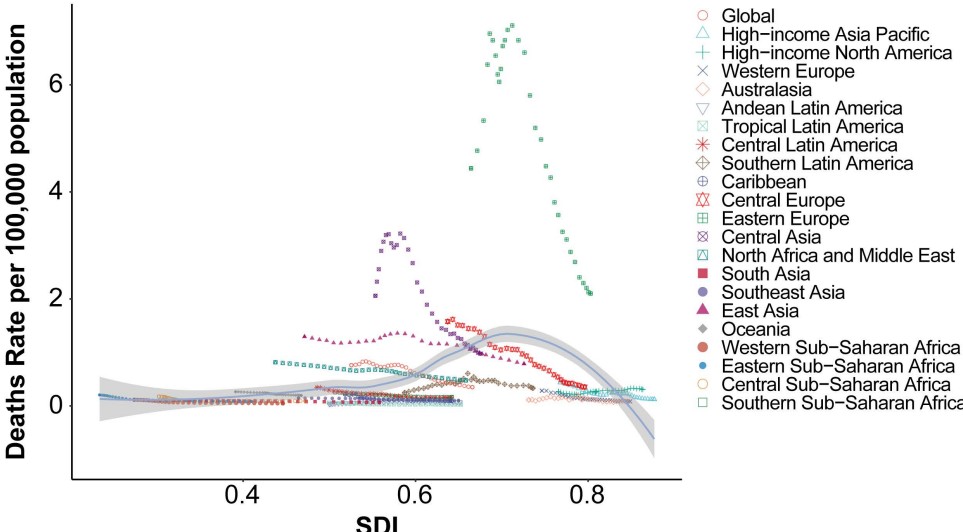

**Fig 5. Association between socio-demographic index and age-standardized mortality rates across 21 global regions.**

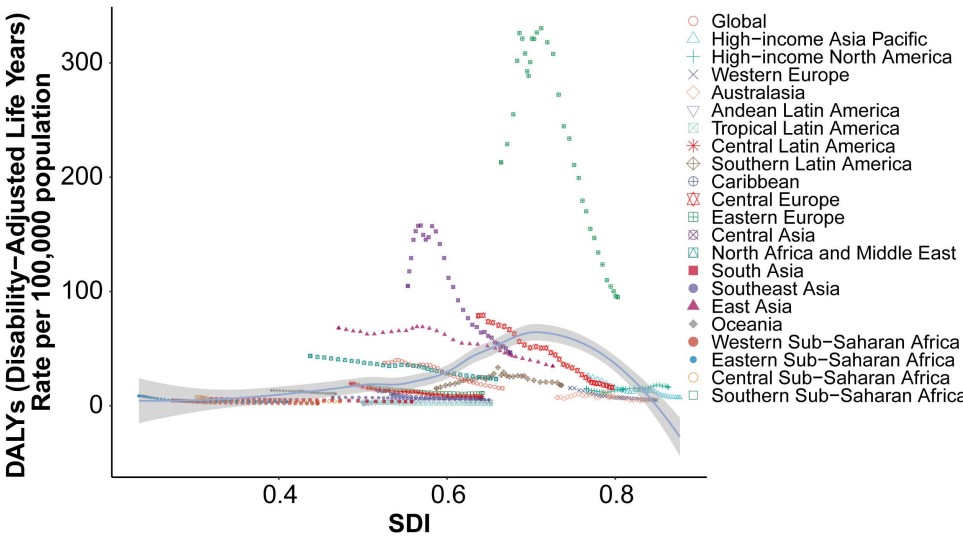

**Fig 6. Association between socio-demographic index and age-standardized DALY rates across 21 global regions.**

### 3.1.3 Regional association between SDI and age-standardized DALY rates.

The relationship between socio-demographic index (SDI) and age-standardized DALY rates for acute carbon monoxide poisoning across 21 global regions showed a moderate positive correlation (Spearman's r = 0.3663, p = 3.671e-25) (Fig 6). The pattern of DALY rates demonstrated marked regional variations across the SDI spectrum.

Eastern Europe exhibited the highest DALY rates, reaching approximately 300 DALYs per 100,000 population at SDI values between 0.6 and 0.7. Central Asia showed the second highest burden, with rates around 150 DALYs per 100,000 population at similar SDI levels.

The global average curve demonstrated a gradual increase with rising SDI values up to approximately 0.6, followed by a decline at higher SDI levels. East Asia showed a distinctive pattern with elevated DALY rates at moderate SDI levels (0.6–0.7), though maintaining lower rates than Eastern Europe.

High-income regions, including Western Europe, High-income North America, and Australasia, showed relatively low DALY rates (below 50 per 100,000 population) despite their high SDI values (0.8–0.9). Most other regions maintained DALY rates below 100 per 100,000 population across the SDI spectrum.

Regions with lower SDI values (0.3–0.5), including South Asia, Southeast Asia, and Sub-Saharan African regions, consistently showed the lowest DALY rates, generally below 25 per 100,000 population, with minimal variation across different SDI values.

## 3.2 National-level analysis

### 3.2.1 National-level analysis of SDI and age-standardized incidence rates.
The relationship between socio-demographic index (SDI) and age-standardized incidence rates (ASR) of acute carbon monoxide poisoning at the national level showed a strong positive correlation (Spearman's $r = 0.7602$, $p < 0.001$) (Fig 7). The analysis across 204 countries revealed distinct patterns of disease burden across the SDI spectrum.

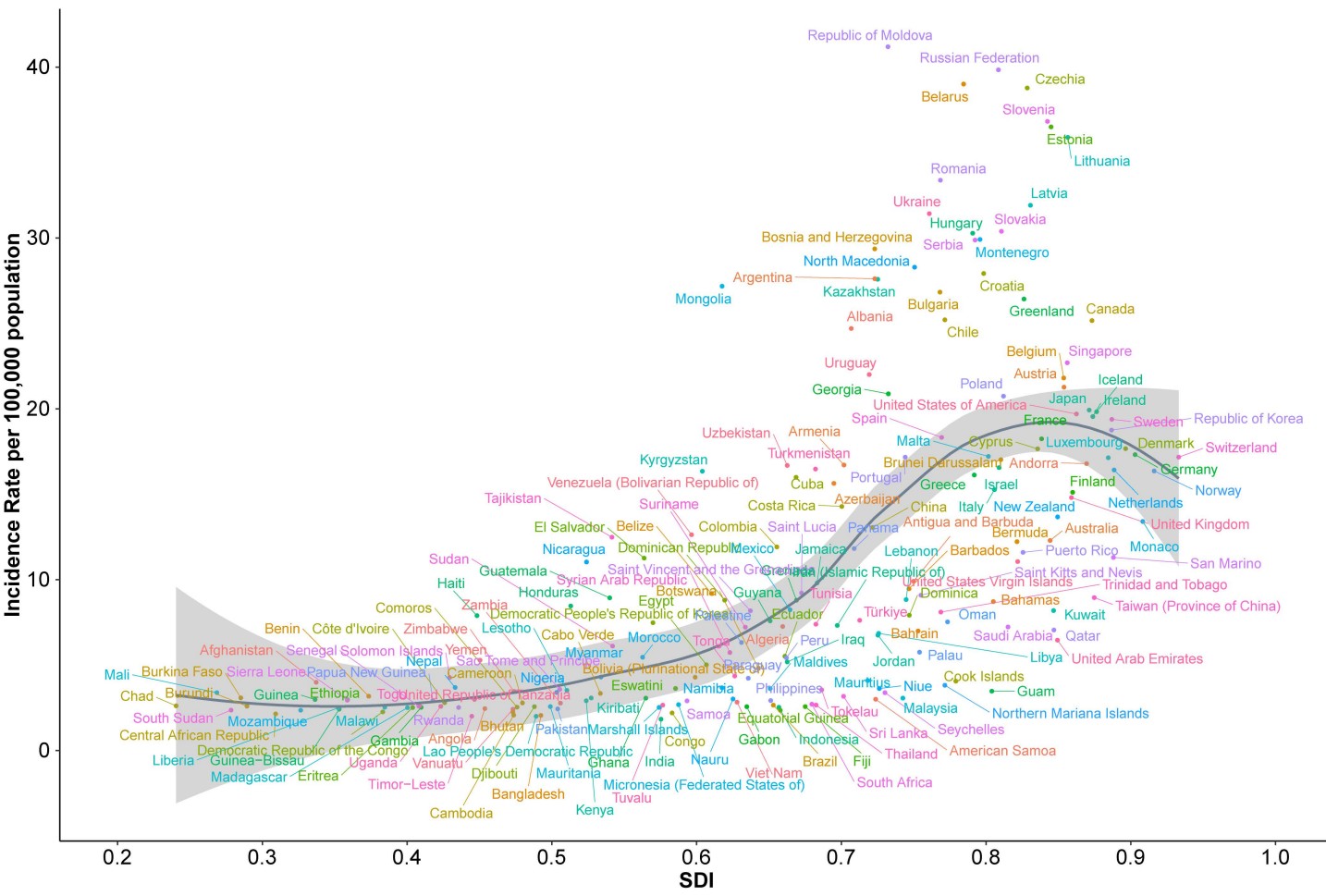

**Fig 7. Association between socio-demographic index and age-standardized incidence rates across 204 countries.**

Countries in Eastern Europe demonstrated the highest incidence rates, with the Republic of Moldova, Russian Federation, and Czechia recording rates above 35 per 100,000 population at SDI values between 0.7 and 0.8. These were followed by Belarus, Slovenia, and Estonia, with rates between 30 and 35 per 100,000 population at similar SDI levels.

A second tier of countries, including Latvia, Hungary, Slovakia, and Bosnia and Herzegovina, showed rates between 25 and 30 per 100,000 population, predominantly clustering at SDI values of 0.7–0.8. Nations with very high SDI values (>0.8), such as Canada, Belgium, and Singapore, generally demonstrated moderate incidence rates between 15 and 25 per 100,000 population.

Countries with lower SDI values (<0.5) consistently showed lower incidence rates, typically below 10 per 100,000 population. These included most nations in Sub-Saharan Africa, South Asia, and parts of Southeast Asia. The overall pattern showed a gradual increase in incidence rates with rising SDI values up to approximately 0.7, after which the relationship became more heterogeneous across countries.

The distribution showed notable regional clustering, with neighboring countries often displaying similar incidence rates despite varying SDI levels, particularly evident in the Eastern European cluster at the higher end of the incidence spectrum and the Sub-Saharan African cluster at the lower end.

**3.2.2 National-level analysis of SDI and age-standardized mortality rates.** The relationship between socio-demographic index (SDI) and age-standardized death rates (ASR) for acute carbon monoxide poisoning across 204 countries showed a weak positive correlation (Spearman's $r = 0.1341$, $p = 5.584e{-}02$) (Fig 8). The country-level analysis revealed distinct mortality patterns across different SDI levels.

The highest mortality rates were observed in Eastern European nations, with the Republic of Moldova and Russian Federation showing rates above 2 per 100,000 population at SDI values between 0.6 and 0.8. Mongolia and Kazakhstan also demonstrated notably high mortality rates, exceeding 1.5 per 100,000 population at moderate SDI levels.

Countries with high SDI values (>0.8), including Western European nations, North America, and high-income Asian countries, generally maintained low mortality rates below 0.5 per 100,000 population. Notable exceptions included Lithuania and Latvia, which showed relatively higher rates despite their high SDI values.

A cluster of countries with low to moderate SDI values (0.3–0.5) exhibited varying mortality rates. Afghanistan showed a distinctly high rate (approximately 1.8 per 100,000) at a low SDI value, while Nepal and Yemen demonstrated moderate rates (0.8–1.0 per 100,000) in similar SDI ranges.

The majority of countries, particularly those with SDI values below 0.6, maintained mortality rates below 0.5 per 100,000 population. This included most nations in Africa, South Asia, and Southeast Asia, showing minimal variation in mortality rates despite their varying SDI levels.

**3.2.3 National-level analysis of SDI and age-standardized DALY rates.** The relationship between socio-demographic index (SDI) and age-standardized DALY rates for acute carbon monoxide poisoning across 204 countries demonstrated a weak positive correlation (Spearman's $r = 0.2233$, $p = 1.356e{-}03$) (Fig 9). The country-level analysis revealed substantial heterogeneity in DALY burden across SDI levels.

Mongolia showed the highest DALY rate, reaching approximately 120 per 100,000 population at a moderate SDI level. This was followed by the Republic of Moldova and Russian Federation, both recording rates above 100 DALYs per 100,000 population at SDI values between 0.6 and 0.8.

A cluster of countries including Kazakhstan, Ukraine, and Belarus demonstrated DALY rates between 60 and 80 per 100,000 population at moderate to high SDI levels (0.6–0.8). Afghanistan showed notably high DALY rates (approximately 85 per 100,000) despite its low SDI value.

Countries with high SDI values (>0.8), including most Western European nations, North America, and high-income Asian countries, generally maintained low DALY rates below 20 per 100,000 population. Notable exceptions included Lithuania and Latvia, which showed moderately elevated rates despite their high SDI values.

**Fig 8. Association between socio-demographic index and age-standardized mortality rates across 204 countries.**

The majority of countries, particularly those with SDI values below 0.6, maintained DALY rates below 40 per 100,000 population. This included most nations in Africa, South Asia, and Southeast Asia, with minimal variation in DALY rates despite their varying SDI levels.

## 4 Global trends and future projections

**4.1 Projected trends in age-standardized incidence rates through 2050.** Global age-standardized incidence rate (ASR) of acute carbon monoxide poisoning is projected to continue its historical decline through 2050 (Fig 10). From an observed rate of approximately 12 per 100,000 population in 1990, the ASR decreased to 8 per 100,000 population by 2020. The forecasting model predicts a further reduction to approximately 5 per 100,000 population by 2050, with widening uncertainty intervals in the long-term projection period.

**4.2 Projected trends in age-standardized mortality rates through 2050.** The global age-standardized mortality rate of acute carbon monoxide poisoning is projected to continue declining through 2050 (Fig 11). The historical trend shows a peak of approximately 0.8 per 100,000 population in the early 1990s, followed by a steady decrease to 0.35 per 100,000 population by 2020. The model forecasts a further reduction to approximately 0.15 per 100,000 population by 2050, with gradually expanding uncertainty intervals in the projection period.

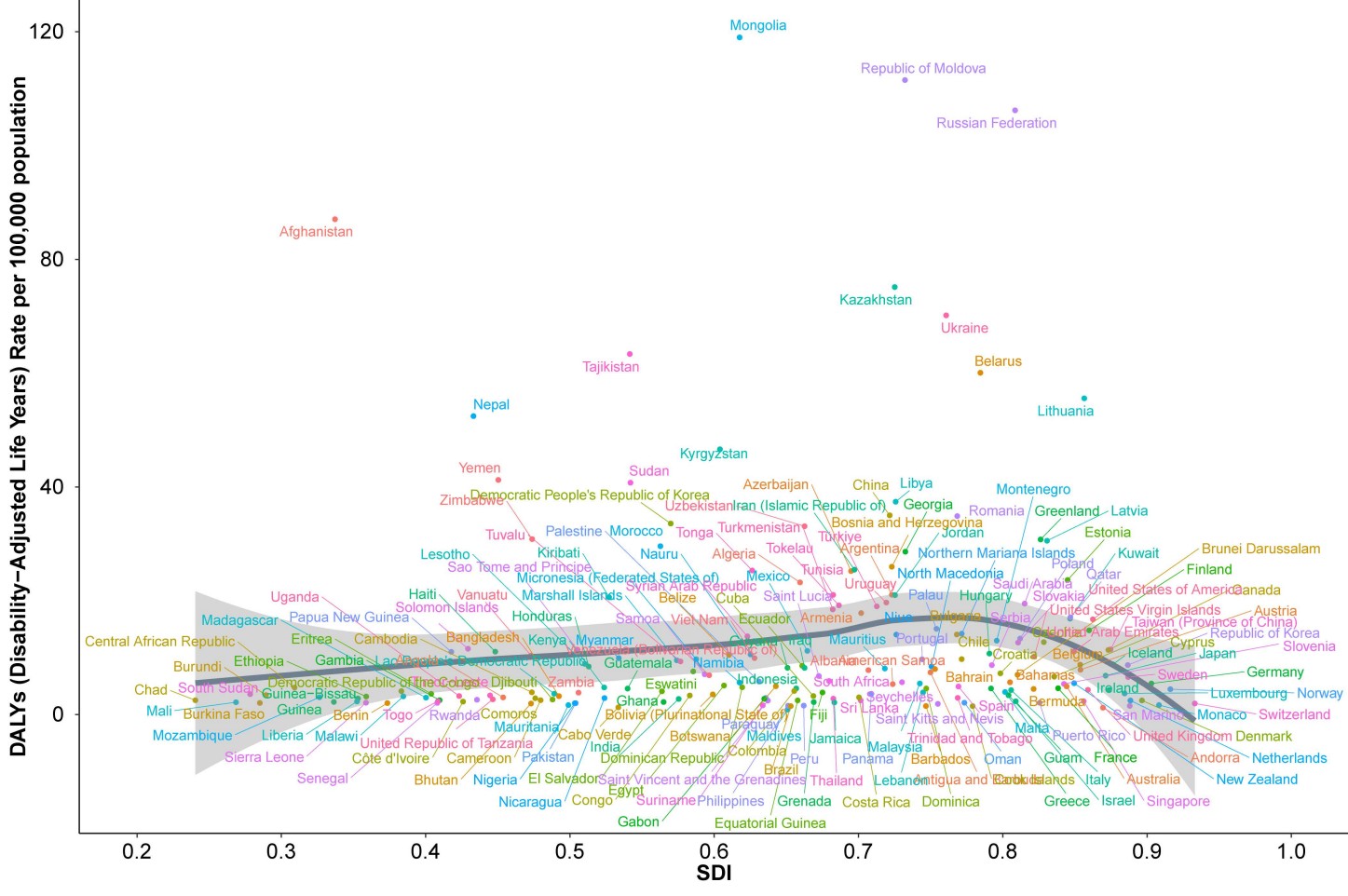

**Fig 9. Association between socio-demographic index and age-standardized DALY rates across 204 countries.**

**4.3 Projected trends in age-standardized DALY rates through 2050.** Global age-standardized DALY rates for acute carbon monoxide poisoning are projected to maintain their downward trend through 2050 (Fig 12). From a peak of approximately 40 DALYs per 100,000 population in the early 1990s, the rate declined to around 15 DALYs per 100,000 population by 2020. The forecasting model predicts a further decrease to approximately 5 DALYs per 100,000 population by 2050, with increasing uncertainty intervals in the latter projection period.

## Discussion

This study provides the first comprehensive analysis of the global burden of carbon monoxide (CO) poisoning, encompassing trends in incidence, mortality, and disability-adjusted life years (DALYs) from 1990 to 2021. Our findings reveal an overall declining trend in the global burden of CO poisoning, albeit with significant variations across geographical regions and socio-demographic index (SDI) levels. These results have important implications for developing targeted prevention strategies.

First, our study demonstrates a substantial global decline in CO poisoning incidence over the study period, reflecting [18], enhanced ventilation facilities [19], and increased public awareness [20]. However, Eastern Europe and High-income Asia Pacific regions maintain relatively high incidence rates, possibly due to high heating demands during winter months and delayed updates of aging heating infrastructure.

Fig 10. Projected trends in age-standardized incidence rates of acute carbon monoxide poisoning through 2050.

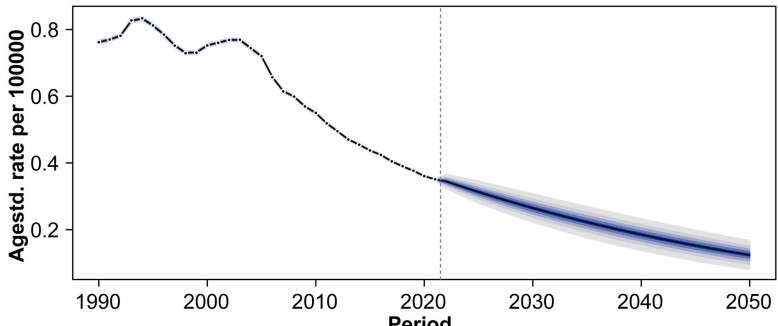

Fig 11. Projected trends in age-standardized mortality rates of acute carbon monoxide poisoning through 2050.

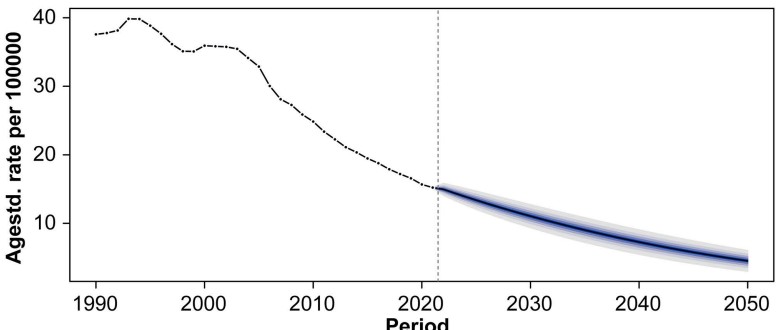

Fig 12. Projected trends in age-standardized DALY rates of acute carbon monoxide poisoning through 2050.

Second, our research reveals a complex relationship between CO poisoning burden and socioeconomic development levels. High-SDI and high-middle SDI regions exhibited the highest incidence rates but relatively lower mortality rates and DALY burdens, suggesting that while these regions report more cases, better medical conditions lead to improved patient outcomes [21,22]. In contrast, low-SDI regions show lower reported incidence rates but potentially suffer from underreporting and limited medical resources, which may result in poorer outcomes [23,24]. This aligns with previous findings [9,11] regarding insufficient acute poisoning treatment capabilities in developing countries.

The geographical distribution analysis reveals significant regional disparities. Eastern Europe consistently maintains the highest burden levels, attributable to its climatic characteristics [25], heating methods, and unique circumstances during socioeconomic transition. The lower reported rates in Africa and Southeast Asia may mask the actual disease burden, indicating a need to strengthen surveillance systems in these regions. Climate factors significantly influence CO poisoning patterns, with colder temperatures increasing heating demands and associated risks [26,27]. Climate change may alter geographic risk patterns, potentially reducing cold-weather exposures while creating new energy-related risks [28]. Behavioral interventions, including public education campaigns, proper equipment maintenance awareness, and widespread CO detector adoption, have proven effective in reducing mortality trends in developed nations [29,30].

Our study is the first to systematically analyze gender disparities, finding that males generally experience higher mortality rates and DALY burdens than females, possibly due to differences in occupational exposure risks and behavioral patterns [31,32].

Several novel findings from our comprehensive analysis warrant detailed discussion. The differential decline rates between incidence (−1.16% annually) and mortality (−2.79% annually) suggest that while exposure prevention efforts are moderately successful, treatment improvements have been more substantial, particularly in high-SDI regions. This pattern indicates that healthcare capacity building may be more achievable than primary prevention in the short term [33]. The inverted U-shaped SDI relationship reveals a critical insight: countries at moderate development levels (SDI 0.6–0.7) face the highest burden, likely reflecting increased fossil fuel use during industrialization without adequate safety infrastructure. This "development trap" suggests that economic growth alone does not guarantee reduced CO poisoning risk without targeted interventions [34]. The pronounced male predominance (2.5-fold higher mortality) extends beyond occupational exposure to include behavioral factors such as risk-taking behaviors and delayed healthcare seeking [35]. Our geographic clustering analysis reveals that neighboring countries often share similar burden patterns regardless of SDI differences, highlighting the importance of regional cooperation in prevention strategies. The persistent Eastern European hotspot, despite three decades of observation, underscores the complex interplay between climate, infrastructure legacy, and socioeconomic transition that requires sustained, multifaceted interventions.

Several limitations should be acknowledged in our study. First, the quality of Global Burden of Disease study data varies across countries, with potential incompleteness in low-income nations. Second, our inability to obtain detailed exposure information and specific causes of death limits concrete guidance for prevention strategies. Third, while our prediction models consider historical trends, they may not fully incorporate potential policy changes and technological advancements. Additionally, our forecasting models may not fully capture emerging variables that could significantly alter future trends. The global transition to clean energy and electric heating systems may accelerate burden decline beyond our projections. Conversely, rapid urbanization, technological innovations in smart monitoring systems, carbon neutrality policies, and climate change-induced extreme weather events introduce variability not reflected in historical trend-based models. These factors suggest our projections represent baseline scenarios rather than definitive forecasts.

These findings support specific evidence-based interventions: mandatory CO detector installation in residential buildings (successful in Canada and United States) [29], national surveillance programs (established in Norway and United Kingdom) [4,36], government-subsidized heating system replacement programs (implemented in Poland and Czech Republic) [37], and strengthened poison control centers following WHO models in developing countries [38,39].

Looking forward, our forecast analysis indicates that the global burden of CO poisoning will continue to decrease through 2050, though regional disparities may persist. This suggests the need for differentiated prevention strategies: high-burden regions should focus on upgrading heating facilities and enforcing safety standards, while low-income regions need strengthened infrastructure and medical treatment capabilities. Future research should explore the underlying causes of regional differences, evaluate the cost-effectiveness of various prevention measures, and establish more comprehensive surveillance systems. Additionally, the promotion of clean energy and development of smart early warning technologies may further reduce the disease burden of CO poisoning.

## Conclusion

This comprehensive global analysis of carbon monoxide poisoning from 1990–2021 reveals significant epidemiological patterns with important public health implications. Our study demonstrates a substantial global decline in CO poisoning burden, with age-standardized incidence rates decreasing by 35.1% from 12.13 to 7.87 per 100,000 population, mortality rates declining by 2.79% annually, and DALYs showing the steepest reduction at 3.18% annually. Despite these improvements, nearly 300,000 new cases and 29,000 deaths occurred globally in 2021, indicating continued substantial burden.

Critical disparities persist across regions and demographics. Eastern Europe maintains the highest burden with incidence rates of 37.98 per 100,000—five times the global average. Males experience 2.5 times higher mortality rates than females across all regions. Most significantly, our analysis reveals a novel inverted U-shaped relationship between socio-demographic development and disease burden, with countries at moderate SDI levels (0.6–0.7) experiencing peak mortality and DALYs burden, while high-SDI regions show higher incidence but dramatically lower case-fatality rates (1.24% vs. 4.26%).

These findings have profound public health significance, demonstrating that economic development alone does not guarantee reduced CO poisoning burden without targeted interventions. The data support differentiated prevention strategies: infrastructure modernization in high-burden regions, strengthened medical capacity in transitioning economies, and enhanced surveillance in low-resource settings. Our projections through 2050 indicate continued global decline but persistent regional disparities, emphasizing the need for sustained, equity-focused interventions. This study provides the first multi-dimensional global assessment of CO poisoning burden and establishes a framework for evidence-based policy making to address this entirely preventable cause of death and disability worldwide.

## Supporting information

**S1 Data. Raw dataset used for Global Burden of Disease (GBD) analysis of carbon monoxide (CO) poisoning.**
(ZIP)

## Author contributions

**Investigation:** Yongai Ling.

**Methodology:** Jiajie Zhou.

**Project administration:** Yongai Ling, Xianwei Xiong, Jiajie Zhou.

**Resources:** Xianwei Xiong.

**Software:** Jiajie Zhou.

**Supervision:** Yongai Ling, Xianwei Xiong, Jiajie Zhou.

**Writing – original draft:** Weiguang Wang.

**Writing – review & editing:** Weiguang Wang.

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
