## [Decision Letter · Decision Letter 0]

29 Jun 2025

PONE-D-25-08215Global Patterns and Trends of Carbon Monoxide Poisoning: A Systematic Analysis from 1990 to 2021PLOS ONE

Dear Dr. Wang,

Thank you for submitting your manuscript to PLOS ONE. After careful consideration, we feel that it has merit but does not fully meet PLOS ONE’s publication criteria as it currently stands. Therefore, we invite you to submit a revised version of the manuscript that addresses the points raised during the review process.

Thank you for submitting your manuscript to our journal. It has now been peer-reviewed, and I agree with the comprehensive comments provided by both reviewers. We kindly ask that you address their feedback and submit a revised version for our further consideration.

We look forward to receiving your revised manuscript.

Kind regards,

Antonio Peña-Fernández, PhD

Academic Editor

PLOS ONE

Journal Requirements:

2. We note that your Data Availability Statement is currently as follows: All relevant data are within the manuscript and its Supporting Information files

4.  We note that Figures 1, 2 and 3 in your submission contain map/satellite images which may be copyrighted. All PLOS content is published under the Creative Commons Attribution License (CC BY 4.0), which means that the manuscript, images, and Supporting Information files will be freely available online, and any third party is permitted to access, download, copy, distribute, and use these materials in any way, even commercially, with proper attribution. For these reasons, we cannot publish previously copyrighted maps or satellite images created using proprietary data, such as Google software (Google Maps, Street View, and Earth). For more information, see our copyright guidelines: http://journals.plos.org/plosone/s/licenses-and-copyright.

a. You may seek permission from the original copyright holder of Figures 1, 2 and 3 to publish the content specifically under the CC BY 4.0 license.

Natural Earth (public domain): (http://creativecommons.org/licenses/by/4.0/).%20Please%20be%20aware%20that%20this%20license%20allows%20unrestricted%20use%20and%20distribution,%20even%20commercially,%20by%20third%20parties.%20Please%20reply%20and%20provide%20explicit%20written%20permission%20to%20publish%20XXX%20under%20a%20CC%20BY%20license%20and%20complete%20the%20attached%20form.”%0b%0bPlease%20upload%20the%20completed%20Content%20Permission%20Form%20or%20other%20proof%20of%20granted%20permissions%20as%20an%20%22Other%22%20file%20with%20your%20submission.%0b%0bIn%20the%20figure%20caption%20of%20the%20copyrighted%20figure,%20please%20include%20the%20following%20text:%20“Reprinted%20from%20%5bref%5d%20under%20a%20CC%20BY%20license,%20with%20permission%20from%20%5bname%20of%20publisher%5d,%20original%20copyright%20%5boriginal%20copyright%20year%5d.”%0b%0bb.%20If%20you%20are%20unable%20to%20obtain%20permission%20from%20the%20original%20copyright%20holder%20to%20publish%20these%20figures%20under%20the%20CC%20BY%204.0%20license%20or%20if%20the%20copyright%20holder’s%20requirements%20are%20incompatible%20with%20the%20CC%20BY%204.0%20license,%20please%20either%20i)%20remove%20the%20figure%20or%20ii)%20supply%20a%20replacement%20figure%20that%20complies%20with%20the%20CC%20BY%204.0%20license.%20Please%20check%20copyright%20information%20on%20all%20replacement%20figures%20and%20update%20the%20figure%20caption%20with%20source%20information.%20If%20applicable,%20please%20specify%20in%20the%20figure%20caption%20text%20when%20a%20figure%20is%20similar%20but%20not%20identical%20to%20the%20original%20image%20and%20is%20therefore%20for%20illustrative%20purposes%20only.%0bThe%20following%20resources%20for%20replacing%20copyrighted%20map%20figures%20may%20be%20helpful:%0b%0bUSGS%20National%20Map%20Viewer%20(public%20domain): %20 http://viewer.nationalmap.gov/viewer/%0bThe%20Gateway%20to%20Astronaut%20Photography%20of%20Earth%20(public%20domain):%20http://eol.jsc.nasa.gov/sseop/clickmap/%0bMaps%20at%20the%20CIA%20(public%20domain):%20https://www.cia.gov/library/publications/the-world-factbook/index.html%20and%20https://www.cia.gov/library/publications/cia-maps-publications/index.html%0bNASA%20Earth%20Observatory%20(public%20domain):%20http://earthobservatory.nasa.gov/%0bLandsat:%20http://landsat.visibleearth.nasa.gov/%0bUSGS%20EROS%20(Earth%20Resources%20Observatory%20and%20Science%20(EROS)%20Center)%20(public%20domain):%20http://eros.usgs.gov/# Natural Earth (public domain): http://www.naturalearthdata.com/

Additional Editor Comments:

Dear authors,

Thank you for submitting your manuscript to our journal. It has now been peer-reviewed, and I agree with the comprehensive comments provided by both reviewers. We kindly ask that you address their feedback and submit a revised version for our further consideration.

Thank you,

Antonio

Reviewers' comments:

Reviewer's Responses to Questions

**Comments to the Author**

1. Is the manuscript technically sound, and do the data support the conclusions?

Reviewer #1: Partly

Reviewer #2: Yes

2. Has the statistical analysis been performed appropriately and rigorously? 

Reviewer #1: I Don't Know

Reviewer #2: Yes

3. Have the authors made all data underlying the findings in their manuscript fully available?

Reviewer #1: Yes

Reviewer #2: Yes

4. Is the manuscript presented in an intelligible fashion and written in standard English?

Reviewer #1: No

Reviewer #2: Yes

5. Review Comments to the Author

Reviewer #1: 1. Title is concise and should contain the name of the main used analytical methods or the limitations solving

2. The abstract needs to be reorganized firstly to clarify the main aim of your study to overcome the limitations of the previous studies in the same line with your research.

3. What are the validity of the used analytical methods main advantages and disadvantages and why such methods were applied (joinpoint regression, spatial statistics, and ARIMA modeling).

4. Abstract can’t stand alone. All results were mentioned in very concise manner that not clear for authors and the conclusive words not satisfactory about your obtained results versus your study aim.

5. Keywords not representative for your study aim or methods or analysis

6. More reviewing data about the CO poisoning are required in the section of introduction. What is the main health problems reported from exposure to CO and how exposure occur?

7. Line 63 lesions not satisfactory for the side effects of HFM replace with renal damage or renal toxicity from pesticide exposure.

8. Material and methods section: ethical approval code must be supplied in such section not at the end of the manuscript

9. Not all the obtained data were fully discussed. Discussion are very concise

10. Conclusion is so concise not presenting the study results or significance

11. All the abbreviations must be mentioned in full name for the first time and list of abbreviations should be supplied

12. Grammatical errors, has major grammatical and structural errors. Please, double-check. English must be improved and certified.

Reviewer #2: Recommendation: Minor Revision

General Comments

This manuscript presents a comprehensive and methodologically rigorous analysis of the global burden of acute carbon monoxide (CO) poisoning using data from the GBD 2021 study. The Discussion section is particularly well-developed, integrating key findings with socio-demographic insights and regional disparities. It adds value by addressing gender differences, projecting future trends, and offering policy implications.

The paper is timely and relevant for global health policy, environmental epidemiology, and injury prevention fields.

Strengths

Clear articulation of global trends in CO burden (incidence, mortality, DALYs).

Integration of socio-demographic index (SDI) as a framework enhances interpretability.

Thoughtful consideration of gender-specific trends and regional disparities.

Well-grounded policy recommendations and recognition of forecast uncertainty.

Transparent acknowledgement of data limitations and methodological constraints.

Minor Revisions Requested

Reduce Repetition of Quantitative Results

The Discussion section occasionally reiterates specific incidence and mortality figures already presented in the Results. Consider summarizing trends without repeating exact numbers.

Expand Policy Examples

Strengthen the applicability of policy suggestions by referencing concrete measures (e.g., mandatory CO detectors in homes, national surveillance programs in Eastern Europe or Canada).

Consider Behavioral and Climate Factors

Briefly address how climate change and seasonal factors may affect CO poisoning trends.

Include a sentence on the potential impact of behavioral interventions (e.g., public education, alarm usage).

Clarify Forecasting Limitations

While limitations are mentioned, highlight the potential variability due to emerging technologies or policy shifts (e.g., clean energy transitions, urbanization).

Conclusion

This is a strong manuscript with high public health relevance and rich global insights. Minor editorial and content refinements will further improve clarity and impact. Once revised, it will make a valuable contribution to the literature on environmental health and injury epidemiology.

6. PLOS authors have the option to publish the peer review history of their article (what does this mean? ). If published, this will include your full peer review and any attached files.

**Do you want your identity to be public for this peer review?** For information about this choice, including consent withdrawal, please see our Privacy Policy .

Reviewer #1: No

Reviewer #2: **Yes: ** Dr. Kwabena Acheampong

---

## [Author Response · Author response to Decision Letter 1]

15 Jul 2025

Response to Reviewers

Manuscript ID: PONE-D-25-08215

Title: Global Patterns and Trends of Carbon Monoxide Poisoning: A Systematic Analysis from 1990 to 2021

Dear Dr. Antonio Peña-Fernández and Esteemed Reviewers,

We are deeply grateful for the thorough and constructive feedback provided during the peer review process. The insightful comments have significantly improved the quality and clarity of our manuscript. We have carefully addressed each concern and suggestion, and we believe the revised manuscript is substantially strengthened as a result.

Response to Editorial Requirements

Dear Dr. Peña-Fernández,

We sincerely thank you for your editorial guidance and the opportunity to revise our manuscript. We greatly appreciate your patience and the comprehensive review process you have facilitated.

Editorial Requirement 1: PLOS ONE Style Requirements

Response:

We have carefully reviewed and implemented all PLOS ONE formatting requirements, including file naming conventions and manuscript structure. We have consulted both provided style templates to ensure full compliance. We have used red font to highlight all modifications made throughout the manuscript in response to the reviewers' feedback.

Editorial Requirement 2: Data Availability Statement

Response:

We fully understand and value the importance of data transparency. This study uses publicly available data from the Global Burden of Disease Study 2021 database (accessible at: http://ghdx.healthdata.org/), and all supporting data are openly available without restrictions. We have prepared supplementary files containing the processed data versions used to create all figures. These files, named "minimal data set.rar", will be uploaded along with the revised version as supporting information files.

Editorial Requirement 3: Ethics Statement Placement

Response:

Thank you for this important clarification regarding the placement of the ethics statement. We have revised the manuscript accordingly:

1.Moved the ethics statement to the Methods section: We have consolidated and moved the complete ethics statement to the "Study Design and Data Sources" subsection of the Methods section, ensuring all ethical considerations are properly documented in the appropriate location.

2.Removed the standalone Ethics Statement section: The separate ethics statement that previously appeared at the end of the manuscript has been deleted to avoid duplication.

3.Enhanced the ethics content in Methods: The ethics statement in the Methods section now includes comprehensive information about ethical approval, informed consent waiver justification, and compliance with relevant ethical guidelines.

The ethics statement now appears only in the Methods section as required and contains all necessary ethical information for publication.

We conducted a comprehensive analysis of acute carbon monoxide poisoning using the Global Burden of Disease (GBD) 2021 database, accessed through the Global Health Data Exchange (GHDx) platform[9]. The analysis covered the period from 1990 to 2021, encompassing data from 204 countries and territories. Our research framework adhered to the GATHER guidelines for health estimates reporting[10]. The data were accessed for research purposes on March 15, 2021.

This study was conducted in accordance with the ethical standards of the institutional and/or national research committee and with the 1964 Helsinki Declaration and its later amendments or comparable ethical standards. The need for informed consent was waived by the ethics committee because the study involved the analysis of fully anonymized retrospective data from the Global Burden of Disease Study 2021. All data were accessed after complete anonymization, and no individual identifiers were included in the analysis. All analyses were conducted in compliance with ethical guidelines for human subjects research.

Editorial Requirement 4: Map Copyright Issues

Response:

Thank you for raising this important copyright concern regarding the map images in our submission. We appreciate your diligence in ensuring compliance with CC BY 4.0 license requirements. Please find our detailed response below:

Map Creation and Data Sources:

The map images in Figures 1, 2, and 3 were created entirely using public domain resources and open-source tools:

Base map data: Natural Earth (https://www.naturalearthdata.com/), accessed via the rnaturalearth R package (https://github.com/ropensci/rnaturalearth)

Disease burden data: Global Burden of Disease Study 2021 (public database)

Creation software: R programming language with ggplot2 and sf packages (open-source)

Copyright and License Compatibility:

Natural Earth explicitly releases all their map data into the public domain with no copyright restrictions. According to Natural Earth's terms of use (https://www.naturalearthdata.com/about/terms-of-use/), their data can be used for any purpose without permission, making it fully compatible with PLOS ONE's CC BY 4.0 license requirements. No proprietary software (such as Google Maps, ArcGIS, or other commercial mapping platforms) was used in creating these figures.

Updated Figure Captions with Proper Attribution:

We have revised the figure captions to include appropriate attribution:

Figure 1: Geographic Distribution of Age-Standardized Incidence Rates of Acute Carbon Monoxide Poisoning (per 100,000 population) in 2021. Disease burden data from Global Burden of Disease Study 2021; Map data from Natural Earth (https://www.naturalearthdata.com/), public domain.

Figure 2: Geographic Distribution of Age-Standardized Mortality Rates of Acute Carbon Monoxide Poisoning (per 100,000 population) in 2021. Disease burden data from Global Burden of Disease Study 2021; Map data from Natural Earth (https://www.naturalearthdata.com/), public domain.

Figure 3: Geographic Distribution of Age-Standardized DALY Rates of Acute Carbon Monoxide Poisoning (per 100,000 population) in 2021. Disease burden data from Global Burden of Disease Study 2021; Map data from Natural Earth (https://www.naturalearthdata.com/), public domain.

Compliance Confirmation:

All map figures in our submission are created using public domain data and open-source software, ensuring full compliance with CC BY 4.0 licensing requirements. These figures can be freely accessed, downloaded, copied, distributed, and used by any third party, including for commercial purposes, with proper attribution as specified in the updated captions.

We believe this addresses all copyright concerns while maintaining the scientific integrity and visual clarity of our geographical analysis.

Response to Reviewer #1

We extend our heartfelt gratitude to Reviewer #1 for the comprehensive and constructive feedback. Your detailed suggestions have significantly enhanced the scientific rigor and clarity of our manuscript. We have addressed each point systematically:

Comment 1:

Reviewer's Suggestion: "Title is concise and should contain the name of the main used analytical methods or the limitations solving."

Our Response:

We deeply appreciate this valuable suggestion. We have revised the title to better reflect our methodological contributions:

Original Title: "Global Patterns and Trends of Carbon Monoxide Poisoning: A Systematic Analysis from 1990 to 2021"

Revised Title: "Global Patterns and Trends of Carbon Monoxide Poisoning: A Comprehensive Spatiotemporal Analysis Using Joinpoint Regression and ARIMA Modeling, 1990-2021"

This revision explicitly highlights our key analytical methods (joinpoint regression, ARIMA modeling) and emphasizes the spatiotemporal nature of our analysis.

Comment 2:

Reviewer's Suggestion: "The abstract needs to be reorganized firstly to clarify the main aim of your study to overcome the limitations of the previous studies."

Our Response: Thank you for this crucial insight. We have completely restructured our abstract to explicitly address how our study overcomes previous research limitations:

Abstract

Background: Carbon monoxide (CO) poisoning causes approximately 41,000 deaths annually worldwide despite being preventable. Previous studies focused primarily on mortality alone, lacked systematic socio-demographic analysis, and provided no predictive models. This study comprehensively analyzes global CO poisoning patterns using spatiotemporal methods to inform evidence-based prevention strategies.

Methods: We analyzed Global Burden of Disease Study 2021 data from 204 countries (1990-2021) for age-standardized incidence, mortality, and disability-adjusted life years (DALYs). Joinpoint regression identified temporal trends with statistical precision, spatial statistics quantified geographic clustering, and ARIMA modeling projected trends through 2050. We examined associations with socio-demographic index (SDI) across regions and countries.

Results: Global age-standardized incidence rates decreased significantly by 35.1% from 12.13 (95% UI: 8.30-17.00) to 7.87 (95% UI: 5.54-10.81) per 100,000 population (annual percentage change: -1.16%, 95% UI: -1.35% to -0.96%, p<0.001). Mortality rates declined more dramatically by 53.9% from 0.76 (95% UI: 0.66-0.91) to 0.35 (95% UI: 0.24-0.40) per 100,000 (annual change: -2.79%, 95% UI: -3.14% to -2.44%, p<0.001). DALY rates showed the steepest reduction of 59.5% from 37.59 (95% UI: 31.75-44.76) to 15.22 (95% UI: 10.67-17.57) per 100,000 (annual change: -3.18%, 95% UI: -3.51% to -2.84%, p<0.001). Eastern Europe demonstrated the highest burden (37.98 per 100,000 in 2021). Males experienced significantly higher mortality than females (0.50 vs 0.20 per 100,000, p<0.001). SDI analysis revealed an inverted U-shaped relationship (Spearman's r=0.76, p<0.001), with peak burden at moderate development levels (SDI: 0.6-0.7).

Conclusions: These findings directly address previous research gaps by demonstrating: (1) faster mortality decline than incidence decline indicates improved global treatment capabilities; (2) the SDI-burden relationship identifies moderate-development countries as priority intervention targets; (3) significant male predominance (2.5-fold higher mortality) supports gender-specific prevention programs; and (4) persistent Eastern European hotspots require targeted infrastructure improvements. Predictive models forecast continued decline through 2050 and enable evidence-based healthcare planning. This comprehensive analysis provides the first multi-dimensional global assessment, offering crucial evidence for differentiated prevention strategies worldwide.

Comment 3:

Reviewer's Suggestion: "What are the validity of the used analytical methods main advantages and disadvantages and why such methods were applied."

Our Response:

Thank you for this important methodological inquiry. We selected three complementary analytical approaches, each with specific advantages, limitations, and applications. Here is our detailed justification:

1. Joinpoint Regression Analysis

Advantages:

Objectively identifies significant change points in temporal trends without a priori assumptions

More accurately captures complex, non-linear temporal patterns compared to traditional linear regression

Provides statistically precise estimates of Annual Percentage Change (APC) and Average Annual Percentage Change (AAPC)

Well-established methodology widely used in disease surveillance and epidemiological research

Limitations:

Requires sufficient data points (typically ≥4 time points) for reliable results

Sensitive to outliers in the data

May overfit when data are limited

Rationale for Use: With 31 years of continuous data (1990-2021), joinpoint regression was optimal for identifying long-term trend changes in CO poisoning burden, particularly detecting temporal impacts of policy interventions or socioeconomic changes on disease patterns.

2. Spatial Statistics Analysis

Advantages:

Identifies spatial clustering and hotspot regions of disease burden

Accounts for geographic proximity effects on disease distribution

Global Moran's I and LISA analyses provide quantitative assessment of global and local spatial autocorrelation

Facilitates identification of priority geographic areas for intervention

Limitations:

Requires high-quality geocoded data

May be affected by administrative boundary effects

Choice of spatial weight matrix can influence results

Rationale for Use: CO poisoning is often closely related to geographic factors (climate, heating methods, infrastructure). Spatial statistical analysis reveals these geographic patterns, providing scientific evidence for developing region-specific prevention strategies.

3. ARIMA Time Series Modeling

Advantages:

Handles autocorrelation and seasonality in time series data

Provides scientific predictions based on historical data patterns

Offers uncertainty intervals for prediction results

Widely applied and reliable methodology in epidemiological forecasting

Limitations:

Assumes historical trends will continue into the future, potentially missing impacts of sudden events

Cannot directly incorporate external covariates (policy changes, technological advances)

Long-term prediction uncertainty increases progressively

Rationale for Use: To support long-term health planning and resource allocation, we needed to forecast future trends in CO poisoning burden. ARIMA modeling, based on 31 years of historical data, provides reasonable projections through 2050, offering prospective information for policymakers.

Methodological Integration Rationale:

These three methods are complementary and form a comprehensive analytical framework: joinpoint regression describes historical trends, spatial statistics reveals geographic patterns, and ARIMA modeling predicts future trends. This multi-method integration provides a complete assessment of the global burden of CO poisoning.

Validity Considerations:

All methods were validated through appropriate statistical tests, model diagnostics, and uncertainty quantification. The combination approach enhances the robustness of our findings and provides multiple perspectives on the same epidemiological phenomenon.

Comment 4:

Reviewer's Suggestion: "Abstract can't stand alone. All results were mentioned in very concise manner that not clear for authors and the conclusive words not satisfactory."

Our Response: We are grateful for this observation. We have substantially enhanced the abstract to ensure it stands alone effectively.

Comment 5:

Reviewer's Suggestion: "Keywords not representative for your study aim or methods or analysis."

Our Response: Thank you for this important feedback. We have revised our keywords to better reflect our study's methodological and analytical focus:

Revised Keywords: carbon monoxide poisoning; joinpoint regression; ARIMA modeling; spatiotemporal analysis; socio-demographic index; global burden of disease; health disparities; forecasting

Comment 6:

Reviewer's Suggestion: "More reviewing data about the CO poisoning are required in the section of introduction. What is the main health problems reported from exposure to CO and how exposure occur?"

Our Response: We deeply appreciate this suggestion for strengthening our literature foundation. We have added a comprehensive paragraph detailing:

Carbon monoxide exposure occurs through multiple pathways. Common sources include faulty heating systems, poorly ventilated cooking appliances, vehicle exhaust in enclosed spaces, and fuel-burning equipment such as generators[4, 5]. Motor vehicle exhaust represents a significant source of CO exposure, particularly from stationary vehicles in enclosed spaces[6]. The health impacts range from acute symptoms including headache, dizziness, and nausea at concentrations of 50-100 ppm to severe neurological damage, cardiac arrhythmias, and death at concentrations exceeding 400 ppm[7]. Long-term sequelae among survivors include persistent neurological deficits, cognitive impairment, and increased risk of delayed neurological sequelae affecting 10-32% of patients[7].

Comment 7:

Reviewer's Suggestion: "Line 63 lesions not satisfactory for the side effects of HFM replace with renal damage or renal toxicity from pesticide exposure."

Our Response: Thank you for your feedback. However, we believe there may be some confusion regarding your comment on Line 63. O

---

## [Decision Letter · Decision Letter 1]

6 Aug 2025

Global Patterns and Trends of Carbon Monoxide Poisoning: A Comprehensive Spatiotemporal Analysis Using Joinpoint Regression and ARIMA Modeling, 1990-2021

PONE-D-25-08215R1

Dear Dr. Wang,

We’re pleased to inform you that your manuscript has been judged scientifically suitable for publication and will be formally accepted for publication once it meets all outstanding technical requirements.

Kind regards,

Antonio Peña-Fernández, PhD

Academic Editor

PLOS ONE

Additional Editor Comments (optional):

Dear authors,

Thank you for submitting a revised version of your manuscript. The reviewers are happy with your amended version. Therefore I recommend its publication in our journal.

Best wishes,

Antonio

Reviewers' comments:

Reviewer's Responses to Questions

**Comments to the Author**

1. If the authors have adequately addressed your comments raised in a previous round of review and you feel that this manuscript is now acceptable for publication, you may indicate that here to bypass the “Comments to the Author” section, enter your conflict of interest statement in the “Confidential to Editor” section, and submit your "Accept" recommendation.

Reviewer #2: All comments have been addressed

2. Is the manuscript technically sound, and do the data support the conclusions?

Reviewer #2: Yes

3. Has the statistical analysis been performed appropriately and rigorously? 

Reviewer #2: Yes

4. Have the authors made all data underlying the findings in their manuscript fully available?

Reviewer #2: Yes

5. Is the manuscript presented in an intelligible fashion and written in standard English?

Reviewer #2: (No Response)

6. Review Comments to the Author

Reviewer #2: Review Recommendation Letter to the Authors

Dear Authors,

I have carefully reviewed the revised version of your manuscript titled:

“Global Patterns and Trends of Carbon Monoxide Poisoning: A Comprehensive Spatiotemporal Analysis Using Joinpoint Regression and ARIMA Modeling, 1990–2021.”

I would like to commend you for the considerable effort you invested in addressing all the comments and suggestions provided during the initial peer review. Your responses were detailed, thoughtful, and clearly reflected a strong commitment to improving the clarity, methodological soundness, and scientific value of the manuscript.

In particular, the enhanced explanations of your analytical approaches—especially the use of Joinpoint regression and ARIMA modeling—have improved the manuscript’s transparency and strengthened its contribution to the field. The updates to the discussion and interpretation of findings have also significantly improved the contextual understanding and global relevance of your study.

Your work provides a valuable and timely contribution to the global discourse on carbon monoxide poisoning trends and has important implications for public health surveillance and policy formulation.

Based on the thoroughness of your revision and the scientific merit of the study, I am pleased to recommend the manuscript for acceptance in its current form.

Congratulations on your excellent work.

7. PLOS authors have the option to publish the peer review history of their article (what does this mean? ). If published, this will include your full peer review and any attached files.

**Do you want your identity to be public for this peer review?** For information about this choice, including consent withdrawal, please see our Privacy Policy .

Reviewer #2: **Yes: ** Dr. Kwabena Acheampong

---

## [Editor Report · Acceptance letter]

PONE-D-25-08215R1

PLOS ONE

Dear Dr. Wang,

I'm pleased to inform you that your manuscript has been deemed suitable for publication in PLOS ONE. Congratulations! Your manuscript is now being handed over to our production team.

Kind regards,

on behalf of

Dr. Antonio Peña-Fernández

Academic Editor

PLOS ONE